# Eukaryotic community composition in the sea surface microlayer across an east-west transect in the Mediterranean Sea

Birthe Zäncker[1,2], Michael Cunliffe[2,3], Anja Engel[1]

[1]GEOMAR, Helmholtz Centre for Ocean Research Kiel, 24105 Kiel, Germany
[2]Marine Biological Association of the UK, Plymouth, PL1 2PB, United Kingdom
[3]School of Biological and Marine Sciences, University of Plymouth, PL4 8AA, United Kingdom

*Correspondence to*: Anja Engel (aengel@geomar.de)

**Abstract.** The sea surface microlayer (SML) represents the boundary layer at the air-sea interface. Microbial eukaryotes in the SML potentially influence air-sea gas exchange directly by taking up and producing gases, and indirectly by excreting and degrading organic matter, which may modify the viscoelastic properties of the SML. However, little is known about the distribution of microbial eukaryotes in the SML. We studied the composition of the microbial community, transparent exopolymer particles and polysaccharides in the SML during the PEACETIME cruise along a west-east transect in the Mediterranean Sea, covering the western basin, Tyrrhenian Sea and Ionian Sea. At the stations located in the Ionian Sea, fungi – likely of continental origin via atmospheric deposition - were found in high relative abundances determined by 18S sequencing efforts, making up a significant proportion of the sequences recovered. At the same time, bacterial and picophytoplankton counts were decreasing from west to east, while transparent exopolymer particle (TEP) abundance and total carbohydrate (TCHO) concentrations remained the same between Mediterranean basins. Thus, the presence of substrates for fungi, such as *Cladosporium* known to take up phytoplankton-derived polysaccharides, in combination with decreased substrate competition by bacteria suggests that fungi could be thriving in the neuston of the Ionian Sea and other low nutrient low chlorophyll (LNLC) regions.

## 1 Introduction

The sea surface microlayer (SML) constitutes the boundary layer between the ocean and the atmosphere (Liss and Duce, 2005; Zhang et al., 2003), and is around 1 to 1000 µm thick (Cunliffe and Murrell, 2009; Liss and Duce, 2005) with distinct physical and chemical properties compared to the underlying water (Cunliffe et al., 2013; Zhang et al., 2003). Due to the prominent position, the SML potentially has a substantial influence on air-sea exchange processes, such as gas transfer and sea spray aerosol formation (Cunliffe et al., 2013; Engel et al., 2017; Freney et al., 2020; Sellegri et al., in prep.).

The microbial food web plays a crucial role in ocean biogeochemistry and has been vastly studied. Despite the fact that microbes in the SML can directly and indirectly influence air-sea gas exchange, few studies have looked at the microbial community composition in the SML, mainly focussing on bacteria (Agogué et al., 2005; Joux et al., 2006; Obernosterer et al., 2008) and less on microbial eukaryotes (Taylor and Cunliffe, 2014). While phytoplankton throughout the water column play

an important role in the ocean as primary producers, phytoneuston in the SML (Apts, 1989; Hardy and Apts, 1984; Naumann, 1917), might have an additional crucial role by impacting air-sea gas exchange (Ploug, 2008; Upstill-Goddard et al., 2003). Early microscopic observations of the SML reported mostly diatoms, dinoflagellates and cyanobacteria (Hardy et al., 1988). More recent studies using 18S rRNA gene sequencing found a decreased protist diversity in the SML compared to underlying

water with chrysophytes and diatoms enriched in the SML (Cunliffe and Murrell, 2010; Taylor and Cunliffe, 2014).

Not only phytoneuston, but also zooneuston, bacterioneuston and myconeuston might influence air-sea gas exchange processes by either parasitizing phytoneuston and thus impacting the primary productivity, or by degrading organic matter available in the SML and producing $CO_2$. While some studies have explored bacterioneuston diversity in the Mediterranean Sea (Agogué et al., 2005; Joux et al., 2006), fungi have not yet been characterized in the SML in this region. Fungi are however abundant

in marine environments (Gladfelter et al., 2019; Grossart et al., 2019; Hassett et al., 2019), living a saprotrophic or parasitic lifestyle and have been found in the Mediterranean Sea before (Garzoli et al., 2015; Gnavi et al., 2017), with the myconeuston studied in other locations (Taylor and Cunliffe, 2014).

Phytoplankton and phytoneuston can release precursors such as carbohydrates which can aggregate and form gelatinous particles such as transparent exopolymer particles (TEP) (Chin et al., 1998; Engel et al., 2004; Verdugo et al., 2004). TEP

contain mainly polysaccharides (Mopper et al., 1995; Passow, 2002), occur ubiquitously in the ocean (Alldredge et al., 1993; Passow, 2002), and are an important structural component of the SML (Wurl and Holmes, 2008). Due to their stickiness TEP can aggregate with other particles (Azetsu-Scott and Passow, 2004; Engel, 2000; Passow and Alldredge, 1995). When the aggregate becomes heavier due to the aggregation with additional particles, it eventually sinks out of the euphotic layer into the deep ocean and may thus play an important role in carbon export (Engel et al., 2004). However, the rate of TEP-related

carbon export does not only depend on its production by phytoplankton, but also on microbial degradation.

Few studies have looked at spatial distribution of the microbial eukaryote communities in the SML and possible environmental drivers of community composition, especially in the open Mediterranean Sea, a characteristic low nutrient low chlorophyll (LNLC) region (Durrieu de Madron et al., 2011). The anti-estuarine circulations at the Strait of Gibraltar and the Straits of Sicily, transport low-nutrient surface waters into the basins, and deeper waters out of the basins, resulting in oligotrophic

conditions in the western and ultra-oligotrophic conditions in the eastern Mediterranean basin (Krom et al., 2004; Mermex Group et al., 2011; Pujo-Pay et al., 2011; Tanhua et al., 2013). The present study focuses on TEP as important structural components of the SML and their precursors, carbohydrates, as well as microbial eukaryotes distribution, focusing on the myconeuston community composition in the SML of the Mediterranean Sea using samples collected during the PEACETIME cruise in May and June 2017.

## 2 Material and methods

### 2.1 Sampling

Samples were collected during the PEACETIME cruise to the Mediterranean Sea onboard the RV *Pourquoi pas?* from the 10th May to the 11th June 2017. A total of 12 stations were sampled from 2.9°E to 19.8°E and 35.5°N to 42.0°N (Fig. 1) collecting water from the SML and the underlying water (ULW) at 20 cm below the SML. SML samples were collected from a zodiac using a glass plate sampler (Cunliffe and Wurl, 2014; Harvey, 1966). The dimensions of the silicate glass plate (50 x 26 cm) resulted in an effective sampling surface area of 2,600 $cm^2$ considering both sides. To avoid contamination during sampling, the zodiac was located in front of the research vessel into the direction of the wind. The glass plate was immersed and withdrawn perpendicular to the sea surface. With a Teflon wiper, SML samples were collected in acid cleaned and rinsed bottles (Cunliffe and Wurl, 2014). A total of app. 1.5 L of SML sample was collected in the course of 1 h. Sampling times are listed in table 1. All sampling equipment was acid-cleaned (10 % HCl), rinsed with Milli-Q and copiously rinsed with seawater from the respective depth once the sampling site was reached. The ULW samples were collected concurrently with two acid-cleaned and MilliQ rinsed glass bottles by immersing the closed bottles and opening them at app. 20 cm.

### 2.2 Gel particle determination

The abundance and area of TEP was measured microscopically (Engel, 2009). The sample volume (10-30 ml) was determined onboard the ship according to the prevailing concentration of TEP. Samples were filtered onto 0.4 µm Nucleopore membranes (Whatman) and stained with 1 ml Alcian Blue solution (0.2 g $l^{-1}$ w/v) for 3 s. Filters were mounted on Cytoclear® slides and stored at -20°C until analysis. Two filters per sample with 30 images each were analyzed using a Zeiss Axio Scope.A1 (Zeiss) and the AxioCam MRc (Zeiss). The pictures with a resolution of 1388 x 1040 pixels were saved using AxioVision LE64 Rel. 4.8 (Zeiss). All particles larger than 0.2 $µm^2$ were analyzed. ImageJ was subsequently used for image analysis (Schneider et al., 2012). 10 ml MilliQ water served as a blank.

### 2.3 Bacterioplankton and bacterioneuston abundance

Bacterial cell numbers were determined from a 2 ml sample fixed with 100 µl glutaraldehyde (GDA, 1 % final concentration). Samples were stored at -20°C and stained with SYBR Green I (Molecular Probes) to determine abundance using a flow cytometer (Becton & Dickinson FACScalibur) with a 488 nm laser. A unique signature in a plot of side scatter (SSC) vs. green fluorescence (FL1) was used to detect bacterial cells. Yellow-green latex beads (Polysciences, 0.5 µm) were used as an internal standard.

### 2.4 Picophytoplankton and picophytoneuston abundance

Picophytoplankton and picophytoneuston cell numbers were determined from a 2 ml sample fixed with 100 µl GDA (1 % final concentration) and stored at -20°C. Samples were filtered through a 50 µm filter and analyzed with a flow cytometer (Becton

90 & Dickinson FACScalibur) using a 488 nm laser and a standard filter set-up. Enumeration of cells was conducted using a high flow rate (app. 39-41 µl min$^{-1}$). The forward or right-angle light scatter (FALS, RALS) as well as the phycoerythrin and chl *a* related fluorescent signal was used to distinguish the cells. Cell counts were analyzed using the CellQuest Pro-Software (BD Biosciences). This method is in accordance with previous method development (Lepesteur et al., 1993).

## 2.5 Total combined carbohydrates

95 Samples (20 ml) for total hydrolysable carbohydrates (TCHO) > 1 kDa were filled into precombusted glass vials (8h, 500°C) and stored at -20°C. In the home lab, high performance anion exchange chromatography with pulsed amperometric detection (HPAEC-PAD) was applied on a Dionex ICS 3000 ion chromatography system (Engel and Händel 2011) for TCHO analysis. Prior to analysis, samples were desalinated with membrane dialysis (1 kDa MWCO, Spectra Por) at 1°C for 5h. Samples were hydrolyzed for 20 h at 100°C with 0.8 M HCl final concentration with subsequent neutralization using acid evaporation (N$_2$,

100 for 5 h at 50°C). Two replicates per TCHO sample were analyzed.

## 2.6 DNA extraction and eukaryote 18S rRNA gene sequencing

400 ml of sample was pre-filtered through a mesh with 100 µm pore size in order to avoid zooplankton being captured on the filters and dominating the retrieved 18S sequences and subsequently filtered onto a Durapore membrane (Millipore, 47 mm, 0.2 µm) and immediately stored at -80°C. In order to improve cell accessibility for the DNA extraction, filters in cryogenic

tubes were immersed in liquid nitrogen and the filter was crushed with a pestle. DNA was extracted according to a modified protocol from Zhou et al. (1996) by Wietz and colleagues (2015). The protocol included bead-beating, phenol-chloroform-isoamyl alcohol purification, isopropanol precipitation and ethanol washing. An additional protein-removal step by salting was used to avoid protein contamination.

Library preparation and sequencing was conducted at the Integrated Microbiome Resource at Dalhousie University, Halifax,

Canada and is described in detail elsewhere (Comeau et al., 2017). Samples were PCR-amplified in two dilutions (1:1 and 1:10) using the 18S rRNA gene primers E572F and E1009R (Comeau et al., 2011). Prior to pooling, samples were cleaned up and normalized using the Invitrogen SequalPrep 96-well Plate kit (Thermo Fisher Scientific). Sequencing was conducted according to Comeau et al. (2017) on an Illumina MiSeq using 300+300 bp paired-end V3 chemistry.

Sequences were processed using the DADA2 pipeline (Callahan et al., 2016) and sequences shorter than 400 bp, longer than

444 bp, with more than 8 homopolymers or any ambiguous bases were discarded. Sequences were aligned with the 18S rRNA gene sequences of the SILVA 132 alignment (Quast et al., 2013). Subsequently, sequences that aligned outside of most of the dataset and chimeras were removed. Sequences were classified using the SILVA 132 database (Quast et al., 2013) and deposited at the European Nucleotide Archive (ENA accession number PRJEB23731). Sequences were not subsampled and sequence numbers per sample ranged from 1063 sequences (S8 SML) to 43,027 sequences (S5 SML), except for PCA, where

all samples were subsampled to 1063 sequences.

**2.7 Statistical analyses**

Statistical analyses and maps were produced using R (R Core Team, 2014) and bathymetry information from NOAA (National Oceanic and Atmospheric Administration). The enrichment factor (EF) was used to compare the concentration of substance A in the SML to the concentration in the ULW and was calculated using the following Eq. (1):


$$EF = \frac{[A]_{SML}}{[A]_{ULW}} \tag{1}$$

Where [A] is the concentration of a parameter in the SML or ULW (World Health Organization, 1995). An EF > 1 indicates enrichment, an EF < 1 indicates depletion and an EF = 1 indicates no change of a phytoplankton genus in the SML compared to the ULW. The significance of difference between the SML and ULW and between the basins of 18S eukaryote sequences and biogeochemical parameters were tested using the Kruskal-Wallis test and PERMANOVA. Correlations were calculated

using Spearman's rank correlation.

**2.8 Data obtained from the ship**

Wind speed, salinity and seawater temperature at 5 m were obtained from the RV *Pourquoi pas?* software. Radiation measurements were obtained with the pyranometer Li-Cor Radiation Sensor (Li-200SZ) measuring wavelengths of 400 to 1100 nm. All parameters were measured every 5 min during the sampling on the zodiac outlined above and the average during

the sampling period was taken for statistical analyses (Table 1).

**3 Results**

**3.1 Microbial eukaryote community composition in the SML and ULW**

The eukaryotic communities in the SML and the ULW were similar (ANOSIM, p=0.039, R=0.1002). The cruise track allowed for sampling in three basins of the Mediterranean Sea: the western basin (Provencal + Algerian basin), the Tyrrhenian Sea and

the Ionian Sea. Looking at the three different basins sampled (Fig. 1), differences were detected in their eukaryotic community composition (Fig. 2). ANOSIM showed that the differences in the eukaryotic community composition were slightly larger across basins than between SML and ULW (p=0.0025, R=0.2263). However, the overall diversity and evenness (based on shannon and pielou indices) were not significantly different between basins (Fig. S1).

16 orders were found in relative abundances over 5 % of the total eukaryotic community in one or more of all 12 stations (Fig.

3). The communities in the SML and ULW at most stations were similar, with Dinophyceae and Syndiniales (Dinoflagellata), and an unidentified Eukaryote class dominating the eukaryotic community. Zooneuston were found in most of the SML samples, but rarely (n=2) in the ULW samples. Zooneuston were comprised of Ploimida (Rotifer), Maxillopoda (Cyclopoida and Calanoida) and Scyphozoa (Semaeostomeae).

Myconeuston and mycoplankton were found in high relative abundances in three ULW samples and in the corresponding SML

samples (S8, ION_2, S7) of the Ionian Sea. At station S7 ULW, fungi made up more than half (54 %) of the total number of

retrieved sequences. The vast majority of fungal amplicon sequence variants (ASVs) (64 out of 69) belonged to Ascomycota and Mucoromycota with the remaining five belonging to the Chytridiomycota (n=3), Basidiomycota and Neocallimastigomycota. Figure 4 displays all fungal ASVs that were recovered throughout the cruise and their relative abundance. It becomes apparent that while fungal ASVs make up a significant amount of sequences in the Ionian Sea (stations
to the right of fig. 4), they were barely detectable at the other stations (p=0.014 for differences on fungal ASV level between basins tested with PERMANOVA).

## 3.2 Concentrations and SML enrichments of microorganisms and organic matter

Bacterial numbers did not show any significant differences between depths. In the SML, bacterial abundances ranged from 2.0 x $10^5$ to 1.0 x $10^6$ cells $ml^{-1}$ with an average of 5.2 x $10^5$ ± 2.3 x $10^5$ cells $ml^{-1}$. In the ULW, bacterial numbers were on average
4.6 x $10^5$ ± 1.5 x $10^5$ cells $ml^{-1}$ (range of 2.2 x $10^5$ – 6.9 x $10^5$ cells $ml^{-1}$) (Fig. 5).

Picophytoneuston (0.2 – 20 µm size range) abundance was on average 3.3 x $10^3$ ± 1.9 x $10^3$ cells $ml^{-1}$ in the SML and picophytoplankton abundance in the ULW was on average 2.3 x $10^3$ ± 1.7 x $10^3$ cells $ml^{-1}$ (range of 1.4 x $10^3$ – 8.5 x $10^3$ cells $ml^{-1}$ in the SML, 9.5 x $10^2$ – 7.1 x $10^3$ cells $ml^{-1}$ in the ULW). Overall, cell counts determined by flow cytometry were significantly higher in the SML than in the ULW (p=0.002, n=12).

TEP concentration was on average 1.4 x $10^7$ ± 9.7 x$10^6$ particles $l^{-1}$ (3.6 x $10^6$ - 3.7 x $10^7$ TEP $l^{-1}$) in the SML. In the ULW, the average TEP concentrations were 3.6 x $10^6$ ± 2.1 x$10^6$ particles $l^{-1}$ (6.8 x $10^5$ - 7.5 x $10^6$ TEP $l^{-1}$) in the ULW. TEP area in the SML was on average 9.7 x $10^7$ ± 1.2 x $10^8$ $mm^2$ $l^{-1}$ (1.5 x $10^7$ and 4.5 x $10^8$ $mm^2$ $l^{-1}$). TEP area was lower in the ULW with an average of 2.3 x $10^7$ ± 1.1 x $10^7$ (2.9 x $10^6$ - 3.9 x $10^7$ $mm^2$ $l^{-1}$). Both TEP abundance and area were significantly enriched in the SML (Fig. 5) with values of p=0.01 and p=0.007, respectively. While irradiation, water temperature and salinity did not
correlate with TEP abundance or area, wind speed did have a significant negative correlation with TEP abundance in the SML ($R^2$ = -0.73) and TEP area in the SML ($R^2$ = -0.75) and the enrichment factor of TEP area ($R^2$ = -0.63).

TCHO concentrations were similar between SML and ULW (Fig. 5), with no significant differences between depths (778 ± 294 nM (562 – 1684 nM) in the SML and 605 ± 97 nM (525 – 885 nM) in the ULW).

## 4 Discussion

### 4.1 Eukaryotic diversity in the surface of the Mediterranean Sea

The eukaryotic community composition between the SML and the ULW only differed slightly, with larger spatial heterogeneity and significant differences between the communities of the Western, Tyrrhenian and Ionian basins. The shannon diversity did not differ significantly between depths or basins, however there was a slight decrease of species richness from west to east (Fig. S1), possibly due to the transition to ultra oligotrophic conditions from west to east, as water exchange with
the Atlantic is most pronounced in the western basin (Reddaway and Bigg, 1996) and organisms have to adapt to a more oligotrophic environment the further east they come.

Looking at the phytoplankton community (Fig. 3), it becomes apparent that no diatoms were present at high relative abundances. In seasonal studies, diatoms have been important during blooms in March and April in the Mediterranean Sea, but later in the year when a stratified water column was established, their importance decreased (Marty et al., 2002). Even though diatoms most likely were not dominant in the samples, finding no diatom orders over 1 % in at least one of the samples might also indicate a bias of the primers used or of the pre-filtration removing larger cells and aggregates. Another point that becomes apparent from Figure 3 is the dominance of dinoflagellate genera. Several studies have shown that dinoflagellates have a large number of 18S rRNA gene copies in comparison to other phytoplankton groups, and therefore the abundance of dinoflagellates in 18S rRNA gene sequencing is often overestimated (Godhe et al., 2008; Guo et al., 2016).

Previous studies suggested various factors that potentially drive the phytoplankton community composition. In addition to buoyancy of cells, radiation, especially in the SML, where often high levels of UV-radiation occur, could potentially cause damage by photoinhibition. Dinoflagellates, one of the dominating phytoplankton groups, can however produce photoprotective compounds, including mycosporine-like amino acids (MAAs) (Carreto et al., 1990; Häder et al., 2007). Even though dinoflagellates can produce MAAs, they can still be inhibited by high UV radiation (Ekelund, 1991). However, looking at the current study, no inhibition by UV radiation can be inferred from the data because phytoplankton were enriched despite high radiation values (e.g. stations S4 and 7) (Table 1). At the same time, TEP were significantly enriched during the sampling campaign while the phytoplankton community did not show significant differences. Previous studies suggested that TEP can protect phytoplankton and bacteria from UV radiation (Elasri and Miller, 1999; Ortega-Retuerta et al., 2009). Further studies would be needed to determine whether TEP production was higher in the SML due to phytoneuston UV protection or whether TEP formation rates were higher in the SML due to wind and wave shear at the surface (Carlson, 1993; Cunliffe et al., 2013).

### 4.2 Fungi in the Ionian Sea

Figures 3 and 4 show the relative prevalence of fungi in the Ionian Sea and their scarce reduced distribution in the western basin and the Tyrrhenian Sea. Most of the ASVs present in the Ionian Sea belonged to Ascomycota and Mucoromycota. With fungi making up a significant part of the apparent eukaryotic community in the Ionian Sea (more than half of the sequences retrieved at one station), the question arises as to what drives the higher fungal relative abundances in this region of the Mediterranean Sea. While fungi, like dinoflagellates and other eukaryotic groups, can have varying amounts of 18S rDNA gene copy numbers, the patchy distribution of fungi found in this study makes a consistent bias unlikely. Marine fungi can live a saprotrophic lifestyle, degrading and recycling high molecular weight organic matter (Chrismas and Cunliffe, 2020; Cunliffe et al., 2017) and potentially competing with functionally similar bacteria. Some marine fungi are also phytoplankton parasites, potentially altering phytoplankton community composition through selective parasitism (Amend et al., 2019; Grossart et al., 2019).. At present, we have a very limited understanding of diversity and functional role of fungi in the SML (myconeuston). One previous study of the coastal myconeuston in the Western English Channel off Plymouth (UK) showed that the SML was dominated by both Ascomycota and Basidiomycota (Taylor and Cunliffe, 2014), compared to Ascomycota dominating in this study.

So far, not many studies have looked at fungi in LNLC regions. A global comparison of fungal distribution (Hassett et al., 2020) has found that fungal diversity determined by amplicon sequencing varies between different oceanic regions with *Exophiala*, belonging to Ascomycota, dominating the Ligurian Sea samples and an unclassified Ascomycota being the most abundant taxon, similar to our study being dominated by Ascomycota.

Not only fungal relative abundances increased in the Ionian Sea, but also ASVs identified as Solanales (Nicotiana) had quite
high relative abundances in the easternmost stations. Since Solanales are land plants, presence of their DNA could suggest a possible strong terrestrial influence on the Ionian Sea, linked to wet or dry deposition that occurred before and/or during our sampling period at ION.. This is also corroborated by air mass trajectory backtracking using the HYSPLIT model (Fig. S3) which showed that aerosols likely were of continental origin (Fu et al., in prep), also confirmed by atmospheric measurements indicating that chemical composition of dry and wet depositions were influenced by Eastern European air masses (Desboeufs
et al., in prep.). Station FAST_2 in the western basin was highly influenced by dust input in the area (Guieu et al., 2020; Tovar-Sánchez et al., 2020). This coincided not only with a high increase in TEP abundance in the SML, but also with a distinct increase in the relative abundance of unidentified dinoflagellates in the SML (Fig. 3). The details of the dust input on the organic matter and microbial community composition in the SML and the ULW are discussed elsewhere (Engel et al., *in prep*). However, figure 4 shows that no fungi were found at station FAST_2 neither in the SML nor in the ULW, showing that dust
input does not necessarily deposit fungi to the surface ocean, which potentially also holds true for the Ionian Sea. In addition, the highest relative abundance of fungi was found in the ULW and not the SML, making a simple atmospheric influence without any subsequent thriving of certain fungal taxa unlikely. In addition to atmospheric inputs, riverine inputs can also influence the Mediterranean Sea (Martin et al., 1989). However, the Ionian Sea itself does not experience vast riverine input and riverine influence is even less pronounced in the open sea, making riverine sources of mycophyta unlikely. Ascomycota
and Mucoromycota have been recovered from a variety of marine environments (Bovio, 2019; Grossart et al., 2019; Hassett et al., 2019), thus implying that they also might be thriving in the SML of the Mediterranean Sea instead of being the result of terrestrial input.

Overall, the most abundant fungal ASV in the Ionian Sea, ASV 8, was identified as belonging to genus *Cladosporium* which has been found in marine environments before (Cunliffe et al., 2017). Another explanation for the high relative abundance of
fungi in the Ionian Sea might be that they are more adapted to dealing with the low nutrient conditions found in the more eastern basin of the Mediterranean Sea.

Bacterial and microalgal numbers determined by flow cytometry decreased significantly from west to east, with bacteria showing the greatest decline (Tovar-Sánchez et al., 2020). While overall microalgal abundances determined by flow cytometry were rather low in the SML and ULW, they were comparable to other studies looking at the phytoplankton abundance in the
SML of the Mediterranean Sea (Joux et al., 2006). The microalgal numbers from 5-200 m (data not shown) were higher than at the air-sea interface. Even though overall bacterial numbers decrease, further molecular analyses would be needed to determine if the bacterial community is changing from west to east and if certain bacterial taxa can benefit from the ultra oligotrophic conditions. At the same time, TCHO and TEP were still abundant in the Ionian Sea, as well as DOC in the SML

and DOC and POC in the ULW which did not show changes between the Ionian Sea and the other basins (Freney et al., 2020; Trueblood et al., 2020). TEP are often enriched in the SML of various oceans (Engel and Galgani, 2016; Jennings et al., 2017; Wurl et al., 2009; Wurl and Holmes, 2008). In previous studies, TEP enrichment was highest over oligotrophic regions (Jennings et al., 2017; Zäncker et al., 2017). This is in good accordance with the present study in the ultra oligotrophic eastern Mediterranean Sea (Durrieu de Madron et al., 2011; Fogg, 1995; Wikner and Hagstrom, 1988) where low picophytoplankton abundances, but high TEP enrichments of 1.1-17.3 were found in the present study. Wind speed correlated negatively with TEP abundance and area in the SML, showing that wind can negatively affect TEP concentrations at the air-sea interface as has been previously suggested (Sun et al., 2018).

Since exchange of water with the Atlantic is mostly pronounced in the western basin and anti-estuarine circulation prevails in the Mediterranean Sea, nutrient limitation increases going eastwards in the Mediterranean Sea. TEP production has been shown to be independent of stoichiometric ratios in the surrounding water before (Corzo et al., 2000). Since especially in the SML, light limitation rarely occurs and TEP might serve as light protection (Elasri and Miller, 1999; Ortega-Retuerta et al., 2009), phytoplankton might still photosynthesize and excrete carbohydrates that assemble to TEP. This would not only explain the lack of difference of TEP abundance between basins, but also TCHO concentrations. However, TCHO could also be produced by cell lysis (due to nutrient depletion) and subsequent release of intracellular compounds into the surrounding water.

TCHO and TEP could therefore provide available substrate and microhabitats for marine fungi with reduced competition by bacteria in the Ionian Sea. *Malassezia* and *Cladosporium* have been shown to assimilate carbon derived from TEP-associated algal polysaccharides in the English Channel (Cunliffe et al., 2017), which highlights that *Cladosporium* and other fungi might be able to make use of the substrate under decreased bacterial competition in the Ionian Sea. In addition, previous studies have shown that the Eastern Mediterranean Sea shows higher concentrations of organic pollutants (Berrojalbiz et al., 2011a, 2011b) and a *Cladosporium* strain has been observed to degrade polycyclic aromatic hydrocarbons (Birolli et al., 2018), highlighting another potential substrate for the fungi detected in the Ionian Sea.

## 5 Conclusions

The present study shows that even though flow cytometry counts suggest that bacteria and picophytoplankton numbers are reducing from west to east of the Mediterranean Sea, organic matter such as microgels and TCHO are still prevalent in surface waters. Our findings from the Ionian Sea suggest that accumulation of organic substrates in the surface under oligotrophic conditions may favour certain taxa such as fungi which can benefit from decreased competition by bacteria. In LNLC regions, where phytoplankton and bacterial counts are typically low, but TEP enrichment is high in the SML might be a specific ecosystem where fungi are able to thrive and to control organic matter degradation.

**Acknowledgements**

We would like to thank the chief scientist, Cécile Guieu and Karine Desboeufs, of the PEACETIME cruise on the RV *Pourquoi*
*pas?*. We would also like to thank the captain and crew of the *Pourquoi pas?* for technical assistance in the field. This work is
a contribution of the PEACETIME project (http://peacetime-project.org), a joint initiative of the MERMEX and ChArMEx
components supported by CNRS-INSU, IFREMER, CEA, and Météo-France as part of the programme MISTRALS
coordinated by INSU (doi: 10.17600/17000300).

We thank Jon Roa for his help in analyzing the total combined carbohydrates and Tania Klüver for analyzing the flow
cytometry cell counts. We would also like to thank ISOS (Kiel, Germany), for funding part of this work with a PhD-
Miniproposal Grant.

**Data availability**

All biogeochemical data will be made available at the French INSU/CNRS LEFE CYBER database (data manager, webmaster:
Catherine Schmechtig). All sequence data is available at the European Nucleotide Archive (ENA accession number
PRJEB23731).

**Author contributions**

BZ, MC and AE wrote the paper and contributed to the data analysis. BZ participated in the sample treatment.

**Competing interests**

The authors declare that they have no conflict of interest

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

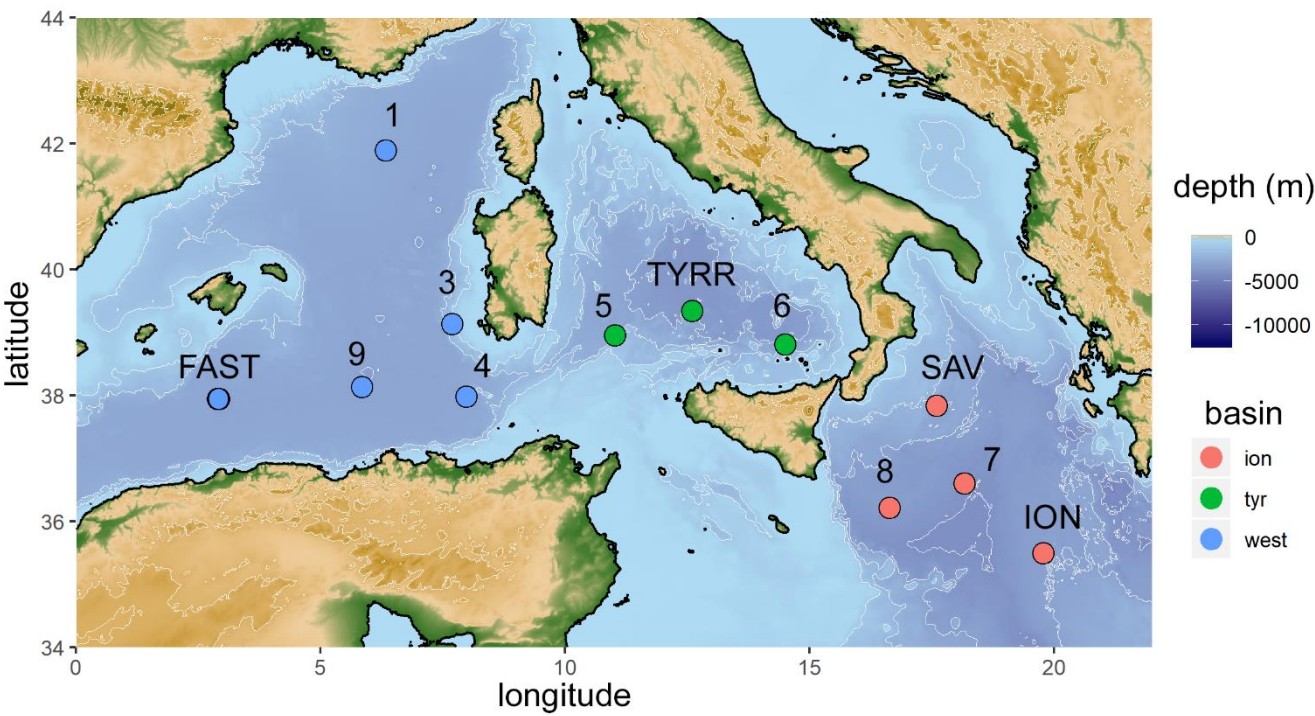

**Figure 1: Map of the stations sampled during the PEACETIME cruise in the Mediterranean Sea in May/June 2017. Stations FAST and TYRR were sampled twice. Colours represent sampled basins (blue: western basin, green: Tyrrhenian Sea, red: Ionian Sea).**

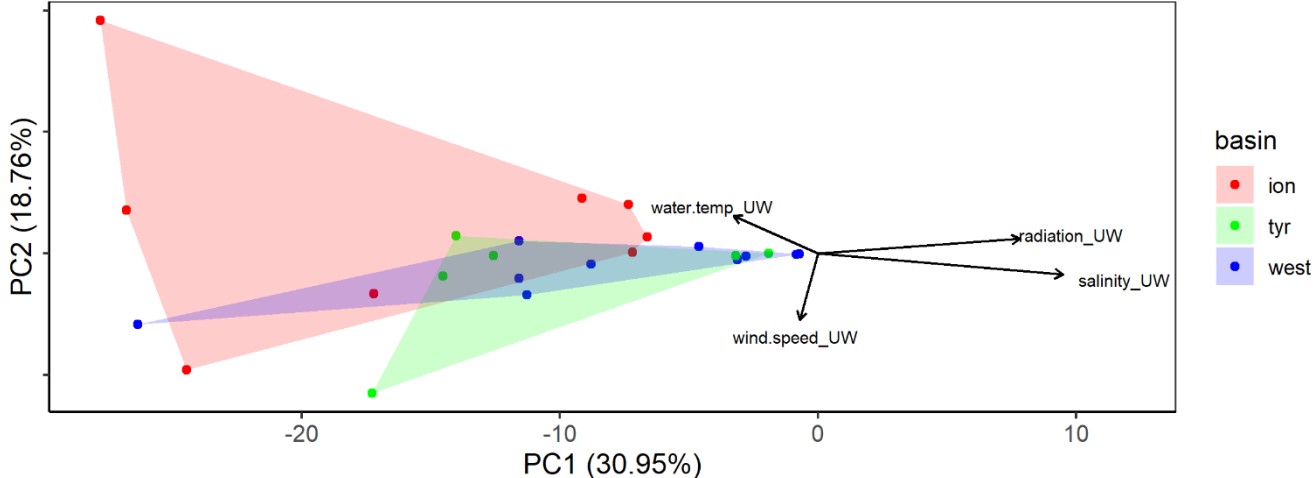

**Figure 2: Principal Component Analyses (PCA) using the eukaryotic community composition on ASV level with environmental factors plotted. Colours distinguish the three different basins sampled (blue: western basin, green: Tyrrhenian Sea, red: Ionian Sea).**

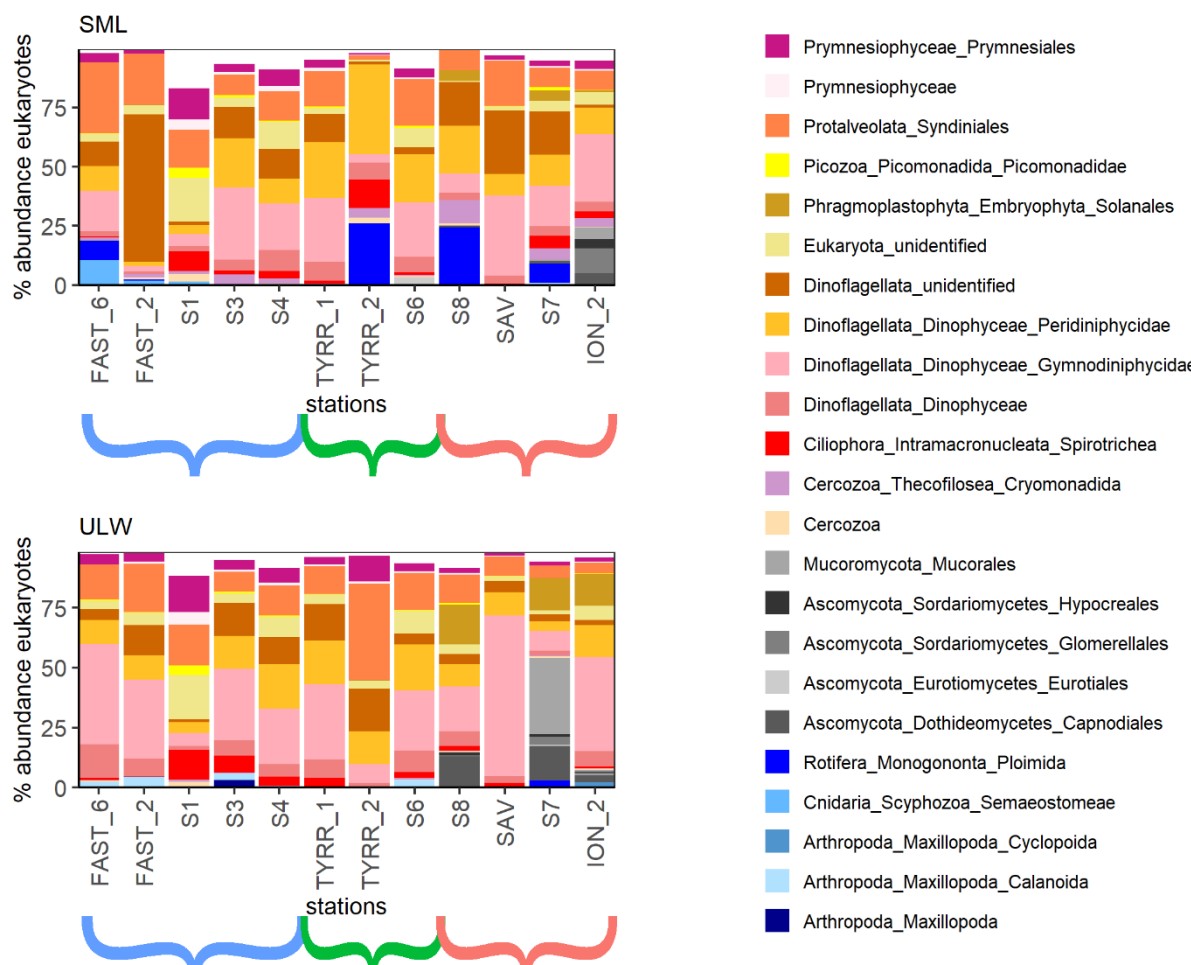

**Figure 3: Eukaryotic community composition on order level (all taxa over 5 percent in at least one of the samples are displayed). Stations ordered from west to east with brackets indicating Mediterranean Sea basins (blue: western basin, green: Tyrrhenian Sea, red: Ionian Sea).**

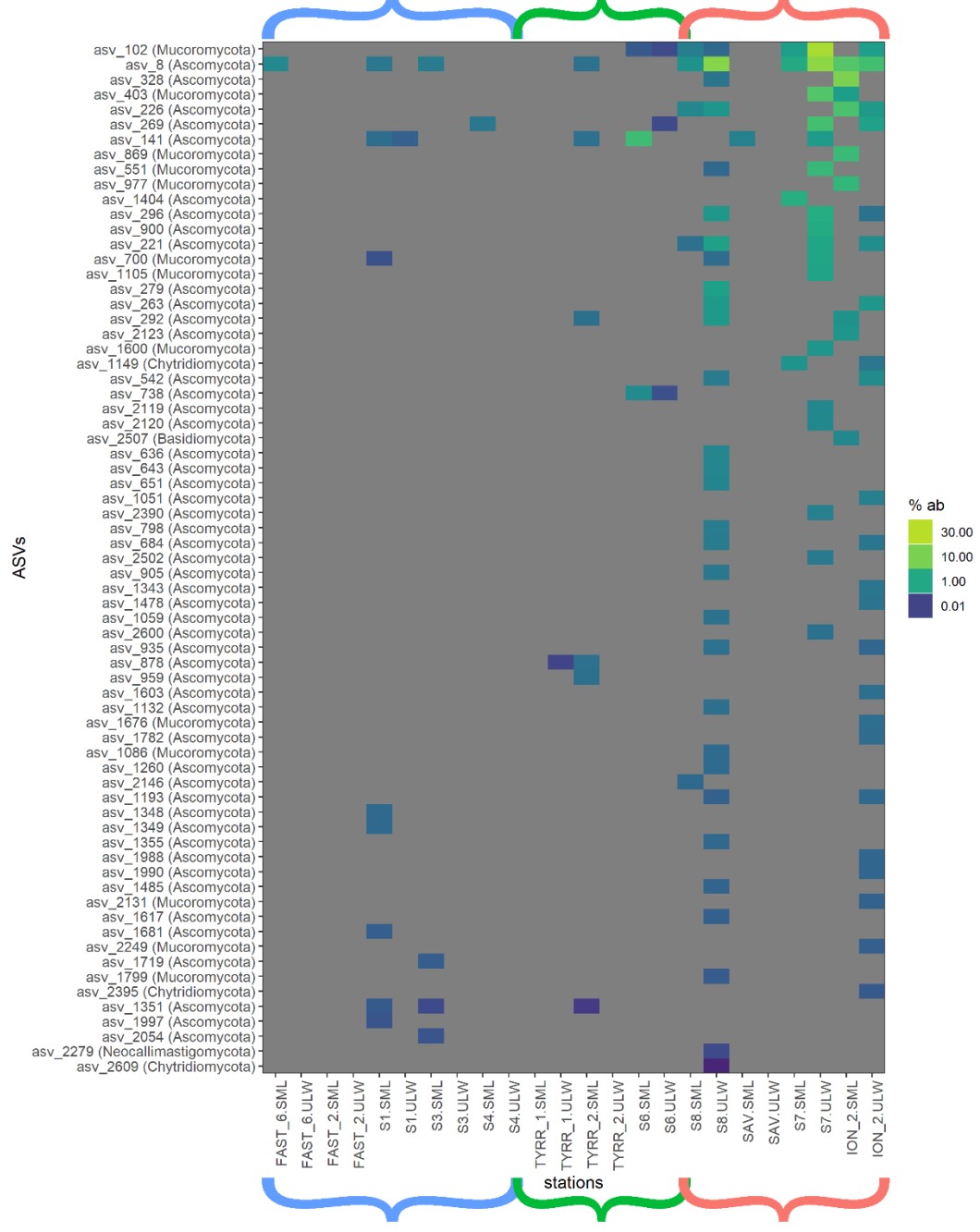

**Figure 4: Heatmap of fungal relative abundances on ASV level throughout the cruise with brackets indicating the Mediterranean Sea basins (blue: western basin, green: Tyrrhenian Sea, red: Ionian Sea). Grey indicates that the ASV was not found in the respective sample.**


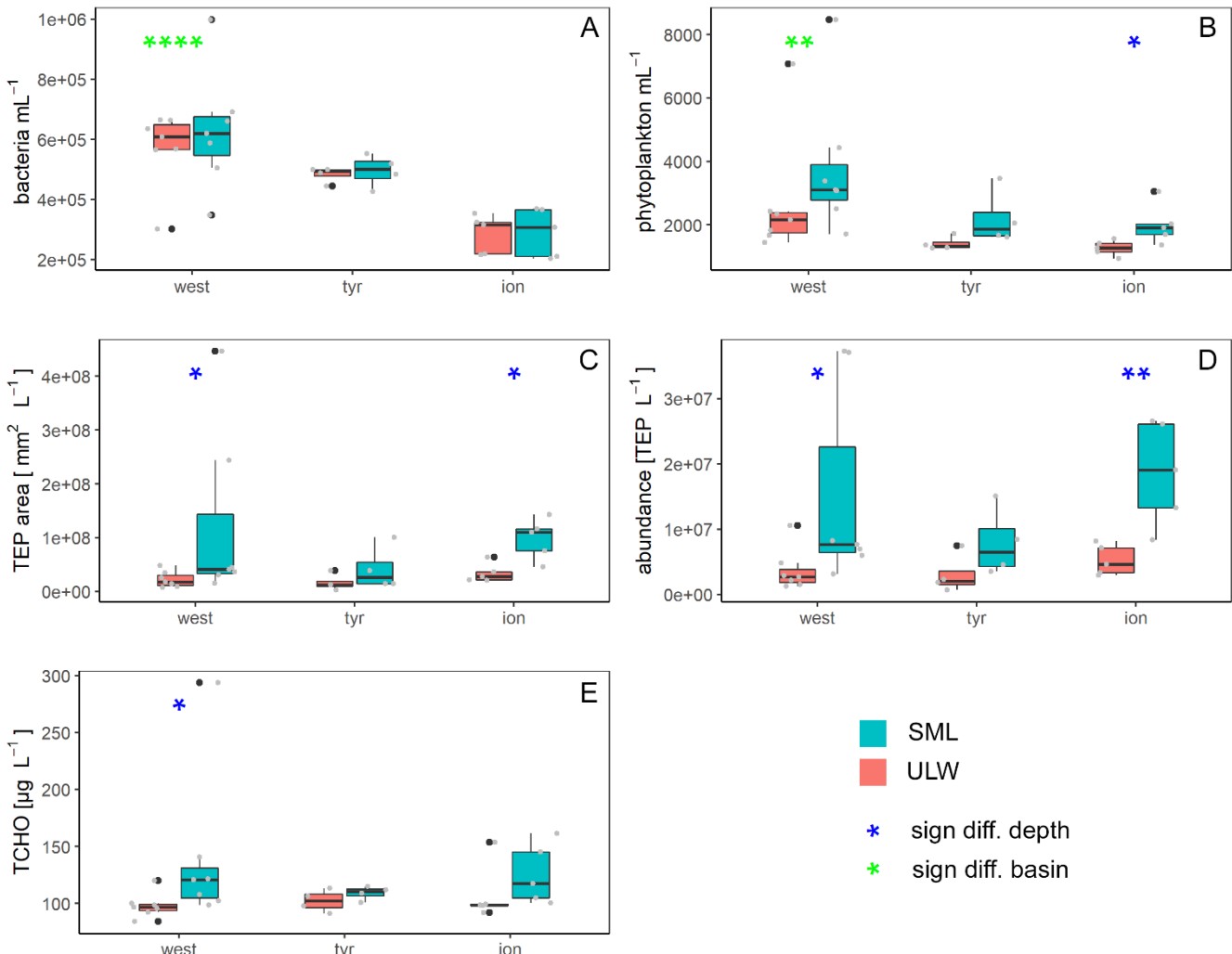

**Figure 5: Abundance of bacteria (A) and phytoplankton (B) as well as area (C) and concentrations (D) of Transparent Exopolymer Particles (TEP) and total carbohydrates (TCHO) (E) across sampled basins in the Mediterranean Sea. Blue stars mark significant SML enrichment/depletion, green stars mark significant differences between the three basins (Kruskall-Wallis tests used for significance levels). Signficance levels: \*: p<0.05, \*\*: p<0.01, \*\*\*: p<0.001, \*\*\*\*: p<0.0001. Black dots mark outliers of the boxplots, grey dots mark the measured values and concentrations.**


**Table 1: Environmental influences at the stations sampled throughout the cruise.**

| station | lat | lon | Local sampling time | wind speed [m s$^{-1}$] | water temp [°C] in 5m | salinity [PSU] in 5m | irradiation [W m$^{-2}$] |
|---|---|---|---|---|---|---|---|
| S1 | 41.8918 | 6.3333 | 15:45 | 9.7 | 16.4 | 38.2 | 1297.8 |
| S3 | 39.1333 | 7.6835 | 10:00 | 2.9 | 18.7 | 37.2 | 2343.2 |
| S4 | 37.9832 | 7.9768 | 10:30 | 3.5 | 19.8 | 37.1 | 2270.2 |
| TYRR_1 | 39.34 | 12.5928 | 11:00 | 3.4 | 20.3 | 37.8 | 2253.1 |
| TYRR_2 | 39.3398 | 12.5928 | 12:30 | 2.5 | 21.1 | 37.7 | 2311.1 |
| S6 | 38.8077 | 14.4997 | 9:00 | 5.2 | 20.4 | 37.4 | 2215.5 |
| SAV | 37.8401 | 18.1658 | 12:00 | 1.5 | 20.1 | 38.5 | |
| S7 | 36.6035 | 18.1658 | 7:00 | 2.5 | 20.8 | 38.5 | 16.8 |
| ION_2 | 35.4892 | 19.7765 | 9:45 | 6.4 | 21.1 | 38.8 | 1235.3 |
| S8 | 36.2103 | 16.631 | 7:45 | 1.9 | 21.2 | 37.9 | 2144.0 |
| FAST_2 | 37.946 | 2.9102 | 8:30 | 3.1 | 21.7 | 36.7 | 627.4 |
| FAST_6 | 37.0466 | 2.9168 | 8:30 | 5.1 | 21.9 | 36.6 | 1787.1 |