# Peer review of "Eukaryotic community composition in the sea surface microlayer across an east-west transect in the Mediterranean Sea"

_Biogeosciences, 2020_

## Referee Comment (RC1) · Anonymous Referee #1 · 26 Jul 2020

Review of manuscript by Zäncker et al. entitled "Eukaryotic community composition in the sea surface microlayer across an east-west transect in the Mediterranean Sea" (bg-2020-249). The study by Zäncker et al. present the spatial distribution of eukaryotic phytoplankton and fungi species in the surface microlayer and the underlying water across different sub-basins at the Mediterranean Sea during summertime. Data show that the SML is a hotspot for different fungi which govern, to some extent, organic matter degradation. Besides, the differences between the SML and the ULW are negligible, and phytoplankton/bacteria show the typical E-W oligotrophic gradient previously reported in numerous studies. Overall, the paper is nicely written, however I think it can be greatly improved. Moreover, I found a few critical points that warrant clarifications;

[Figure]

mostly in the sample's collection (e.g., DNA extraction and different collection hours) and preservation (e.g., flow-cytometry analyses). To conclude, I think the paper should undergo a major revision before I can recommend its publication in Biogeoscience. General comments/suggestions • Comparison of the SML (and ULW) properties between basins/station may be problematic as it seems that the samples were collected in different hours of the day (e.g., station S7 vs. S6). Different collection hours may affect the phytoplankton composition through top-down interactions (i.e., daily migration of zooplankton). The authors should discuss this possible bias. • I suggest adding a short paragraph in the introduction describing the Mediterranean's general west to east anti-estuarine circulation and the trophic gradient it generates (i.e., easternmost stations are 'more oligotrophic' than the western stations, etc.). There is also a N-S trophic gradient that may be relevant to this study (and is not discussed at all in the results or discussion sections). This is the rational for taking samples in different basins across the Mediterranean, representing different oligotrophic characteristics... • Phytoplankton abundance measurements may be underestimated due to wrong preservation of the samples. Freezing the seawater samples in -20 $^\circ$C rather than in liquid nitrogen and then -80 $^\circ$C slowly generates ice crystals that may break some of the cells, and thus result in underestimation of the actual counts. Indeed, the pico-phytoplankton cell abundances presented in Figure 5 (and corresponding text) are 1-2 orders of magnitude lower than usually reported in the Mediterranean Sea ($\sim$104-105 cells/ml). • In section 2.7 you describe how you calculated the 'enrichment factors' between the SML and ULW, however this data is not presented in the manuscript (but only used as a correlation variable). I suggest adding a table with the EF values (and whether the differences were significant or not). It will greatly help the reader to understand the differences between the two water layers. • Figure 5 should be revised. Briefly, there's seems to be problems in the units used for TEP (area and concentration), the panels are not numbered which makes the reading more difficult to follow, the dot's color-code is unclear etc. Please see more details in the table below. Further, I suggest adding to the supporting information a few microscopic images showing example TEP area. • By pre-filtering the seawater onto 100 $\mu$m mesh (line 96), you may have removed some fungi and large-size diatoms/dinoflagellates (as indicated in lines 175-177). Please justify this pre-filtration step. • I think the discussion should be elaborated. For example:  You should discuss why you don't see any differences in the eukaryotic diversity between the SML and underlying water in all sites (it's not the organic matter. . .). Given the lack of (spatial) correlation between phytoplankton and microgels/ TCHO may infer that these organic matter may be refractory in the eastern basin compared to the western basin, or that phytoplankton/bacteria are outcompeted for these 'goods' (perhaps the fungi?).  I suggest you discuss how fungi may inter-act with phytoplankton and bacteria in marine LNLC environments. Do they utilize the same nutrients (thereby competing with the microbes)? Did you find any toxic fungi in the different layers? Can you say anything about the role of fungi in the SML and ULW's food web?  Please provide information on fungi biomass/ activity and diversity in other LNLC regions. Do you expect that fungi be more important in oligotrophic vs. meso-/eutrophic marine environments? Does your findings comparable to these other sites? Minor comments/suggestions Section 2.1 It is unclear how much water were collected in each station, what were the collection hours (day vs. night. . .), and how much time it took to collect it. Seems that for all analyses the authors needed ∼0.5 L from the SML, which is a lot when using the glass plate approach (Harvey, 1966). . .

More importantly, if samples were collected in different hours of the day (e.g., S7 vs. S3 based on irradiance presented in Table 1), this might affect the microbial communities in the SML and ULW through daily migration of zooplankton and thus gazing. This issue can affect the abundance/diversity of the eukaryotic microbes in both water layers. Please provide the information and discussion where appropriate.

Did you use a mechanic instrument (that can also control the sampling rate; ∼17 cm s-1)? If so, an image showing this instrumentation may be a nice addition (especially given that the link you provided of Cunliffe and Wurl, 2014 does not work. . .). Line 57 Pourquoi pas? or Pourquoi pas Line 68 ". . .The abundance and area of TEP and was

measured microscopically. . ." (removed "and") Line 77 (ditto line 83) Freezing seawater samples in -20 °C without pre-freezing it in liquid nitrogen may result in cell lose. The slow freezing at -20 °C creates ice crystals which results in cell breakdown, thus leading to underestimation of pico-phytoplankton/bacterial abundances. Indeed – your pico-phytoplankton cell abundance (e.g., Figure 5) is low by 1-2 orders of magnitude relative to previous studies from the Mediterranean surface water. Do the numbers presented in Figure 5 only show the eukaryotic algae (i.e., without the cyanobacteria)? Abstract, Line 82 and throughout Your flow-cytometry analyses enabled you to enumerate pico/nano-phytoplankton and not the total phytoplankton fraction which is also comprised of larger algal communities (large diatoms, dinoflagellates etc.). What is the cell-size range of the flow-cytometer you used? Usually, to get total "phytoplankton" you should have measured chlorophyll.a and /or run complimentary microscope analyses. This is especially important given that the SML is rich in large-size phytoplankton (Hardyet al., 1988). I suggest changing the term "phytoplankton" to "pico-phytoplankton" throughout.

BTW – Did you see any differences (SML vs. underlying water and between basins) in the pico-phytoplankton communities (e.g., Prochlorococcus:Synechococcus ratio, prokaryotes:eukaryotes ratio)? Paragraph in lines 42-49 I suggest to remove from the introduction (and maybe move it to the discussion?). I'm aware you tried to describe what is TEP, but it has little connection the way it's written with the SML's background. Maybe adding a sentence saying that TEP prevalent the SML. . . etc. Line 50 "looked at the spatial distribution. . ." (add "the") Figure 5 • Please number the different panels (A, B. . .) and revise the legends accordingly ("Abundance of bacteria (A), pico-phytoplankton (B), TEP area (C). . ."). • What's the difference between the gray and black dots (different cruises?)? • TEP area – I don't understand the units. What is mm L-1 (ditto in the text)? I suggest adding a figure in the SI (or atl least in your reply) explaining this. • TEP concentration – I don't understand the units. what is TEP per L-1 (ditto in the text)? Do you mean $\mu$g GX L-1 (if so, there's something off with the numbers). Line 96 and lines 175-177 Please justify why you

used pre-filtration for the DNA extractions. By doing so, you may have taken out fungi's mycelium as well as large-size diatoms/dinoflagellates (that are often found in the western basin water as indicated in lines 175-177). You may have also taken out TEP with its rich microbiome (algae, bacteria and fungi). Section 4.1 You should discuss why you didn't see any differences between the SML and underlying water in all sites, while chemically-wise (total carbohydrates and TEP) you found significant differences. Currently, the discussion in this section is a bit weak. Lines 195-197 There's also a possibility it's a contamination... Did you run blank filters? You can also look at the air-mass backward trajectories (https://ready.arl.noaa.gov/HYSPLIT_traj.php) and see where aerosols came from the day before you sampled there, namely if you received any terrestrial origin particles. Line 217 "very oligotrophic" (instead of "veryoligotrophic"). BTW- I suggest saying 'ultra-oligotrophic'. Lines 218-219 Please cite a reference to back up this statement.

Please also note the supplement to this comment:
https://www.biogeosciences-discuss.net/bg-2020-249/bg-2020-249-RC1-supplement.pdf

---

## Referee Comment (RC2) · Anonymous Referee #2 · 16 Sep 2020

Review of "Eukaryotic community composition in the sea surface microlayer across an east-west transect in the Mediterranean Sea"

This is an interesting work reporting original data on the abundance of eukaryotes and TEPs in a East-West transect in the Mediterranean. The data set seems of good quality, even though the depth of the discussion could be improved in some sections. I recommend the following modifications:

- Line 28. There are some reports on bacteria in the SML, also some recent work using 16S. I suggest this could be cited here.

- Lines 43-49. This paragraph is un-related with the rest of the introduction and the title

of the manuscript. It could be removed from the introduction.

- Line 53. It is not clear to me to which degree the organic matter in the SML is related to atmospheric inputs, it may be driven by partitioning of surfactant-like chemicals from underlying waters or exudates from microbes. In addition, later in the text it is said that the atmospheric influence is covered in another manuscript in preparation.

- Lines 63. I find it difficult to believe that such a control of this rate can be achieve, especially on a moving zodiac.

- Line 65. How was it collected?

- Volume of SML and ULW collected?

- Line 131. Is this the result of a spearman correlation? R=0.1 means R2=0.01. Even though p<0.05, I don't think this shows a high similarity. I think it needs an explanation of which data was used for this similarity measure.

- ASV is never defined in the manuscript, I guess it is "amplicon sequence variant", but a definition, and probably an explanation, is needed in methods or the first time it appears.

- Were differences between SML and ULW tested with a paired test?

- Generally, the EF could be correlated with environmental variables such as wind speed.

- Was DOC measured? This could influence the east-west differences, as well their composition.

- Line 153. Rewrite. . .

- Line 176. Couldn't diatoms and other phytoplankton groups be very affected by UV radiation?, ok, this is commented later, but then the enrichment of phytoplankton can be derived by physical processes (buoyancy and fractionation at surface due to surface

tension related issues), then this would be independent of radiation.

- Note that the SML is generally enriched in hydrophobic and surfactant-like chemicals as many anthropogenic compounds. This has been described for PAHs, alkanes, PCBs, Perfluoralkyl substance, etc in the Mediterranean and elsewhere. This could also have an influence on the east-west differences, as concentrations of POPs in biota are higher in oligotrophic regions due to a complex interplay of factors (Berrojalbiz et al. 2011, Morales et al. 2015, González-Gaya et al. 2019). For example, for bacteria, it has been shown that the SML is especially enriched in those taxa having the potential to degrade pollutants (Martinez-Varela et al. 2020). Even though, organic pollutants were probably not measured in this cruise, it could be another factor to take into account or comment shortly. Fungi are known as being very efficient degrading persistent pollutants.

- Line 231. Here and in other parts of the manuscripts. Bacteria was determined by bacterial counts (abundances), but this is a very limited information for this types of statements, as even bacterial abundance decrease, the abundance of some key taxa may increase.

---

## Referee Comment (RC3) · Anonymous Referee #3 · 1 Oct 2020

General comments: The MS by Zäncker and colleagues reports data on autotrophic and heterotrophic microbial cell abundances, TEP, carbohydrates, and 18S sequences from water collected in the sea surface microlayer (SML) and in underlying waters (ULW) in different basins in the Mediterranean Sea. The work presents a partial description of the biological and chemical characterization of the SML and ULW. However, the way the data are presented and in particular discussed leaves the question on the authors' specific aim(s) open. The question that arises is 'What is the link between TEP, carbohydrates and microbial communities?' The authors provide no rationale for combining these specific results in one MS. Further, the results are compared between SML and UW as well as among basins, which adds another level of complexity. The

discussion of the data (in particular in the context of atmospheric deposition) is difficult to follow. This MS is a contribution to the Special Issue of the PEACETIME project. The characteristics of the SML will certainly provide important insights to the overall project. I consider, however, that the MS cannot be accepted in its present form, but needs major revisions. I advice the authors to re-consider their main objective(s) and to present only the appropriate data. My further suggestion is to re-construct the discussion in a way that it focuses on the data presented in this MS. An original finding of the study is the high relative abundance of fungi sequences in the Thyrennian Sea, both in the SML and ULW. One possibility would be to focus the MS on eukaryotic diversity and fungi in particular.

Specific comments: Abstract: Line 10-11: One understands that this main objective of the work, but it is not focus of the following sections. Methods: Line 89: If I understand correctly, the 20 mL samples collected onboard and frozen (-20°) were not ultra filtered. The ultrafiltration step was done in the lab. I suggest to clarify this. Results: Line 150: Please refer to Fig. 5 in the text Line 153-155 and legend of Fig. 5: Flow cytometry was used in the present study to determine phytoplankton abundance in the SML and in ULW. I suggest the authors clarify here the size fraction of the organisms that can be determined by flow cytometry (i.e. generally up to 20 $\mu$m). Any larger phytoplankton are not included in their counts. Line 160: should be Fig. 5 Discussion: Line 173: Sequencing data provide information on the relative abundance of a given taxonomic unit, but no absolute values. I suggest re-writing the sentence accordingly. Line 191: In the previous paragraph the authors discuss the potential biases of sequencing data due to differences in gene copy numbers, and I totally agree. How would this impact their observations on fungi sequences? I suggest the authors include a short description of what is known on fungi copy numbers and whether this could have led to a potential overestimation in their data set. Line 194 and elsewhere: Please apply the term 'relative abundance' instead of 'abundance' or 'amount' (line 203) Line 198: What is the rationale for the conclusion that dust and rain lead to an increase in TEP and in unidentified dinoflagellates in the SML? Even if another MS on this issue is in

preparation, a more information is required here for an appropriate discussion. What is meant by 'previous to the research campaign'? A few days or weeks? Where is this shown in Fig. 3? Line 201: As mentioned above, this discussion does not refer to any data presented in this MS, and thus confusing. Line 202-203: It seems the authors contradict their statements above. Please clarify.

―――――――――――――――――――――

---

## Author Response (AR2)

Comments to the Author: Dear author,
the text and figure legends are not of sufficient quality and need to be
improved. Please find my suggestions in the annotated manuscript. I am
please to accept publication of your manuscript pending these minor
revisions.
Sincerely,
Christine Klaas

We thank the editor for her helpful comments and have incorporated the minor text changes into
the manuscript (please see tracked manuscript version for details).

---

## Author Response (AR3)

**Reply reviewer #1**

Review of manuscript by Zäncker et al. entitled "*Eukaryotic community composition in the sea surface microlayer across an east-west transect in the Mediterranean Sea*" (bg-2020-249).

The study by Zäncker et al. present the spatial distribution of eukaryotic phytoplankton and fungi species in the surface microlayer and the underlying water across different sub-basins at the Mediterranean Sea during summertime. Data show that the SML is a hotspot for different fungi which govern, to some extent, organic matter degradation. Besides, the differences between the SML and the ULW are negligible, and phytoplankton/bacteria show the typical E-W oligotrophic gradient previously reported in numerous studies.

Overall, the paper is nicely written, however I think it can be greatly improved. Moreover, I found a few critical points that warrant clarifications; mostly in the sample's collection (e.g., DNA extraction and different collection hours) and preservation (e.g., flow-cytometry analyses). To conclude, I think the paper should undergo a major revision before I can recommend its publication in *Biogeoscience*.

General comments/suggestions

We thank the reviewer for the suggestions and helpful comments which we have addressed below.

□ Comparison of the SML (and ULW) properties between basins/station may be problematic as it seems that the samples were collected in different hours of the day (e.g., station S7 vs. S6). Different collection hours may affect the phytoplankton composition through top-down interactions (i.e., daily migration of zooplankton). The authors should discuss this possible bias.

As mentioned in the results section 3.2, influence of irradiation was tested with the two stations with by far the lowest radiation representing the early morning sampling, but no significant correlations were found, making day/night sampling effects unlikely.

□ I suggest adding a short paragraph in the introduction describing the Mediterranean's general west to east anti-estuarine circulation and the trophic gradient it generates (i.e., easternmost stations are 'more oligotrophic' than the western stations, etc.). There is also a N-S trophic gradient that may be relevant to this study (and is not discussed at all in the results or discussion sections). This is the rational for taking samples in different basins across the Mediterranean, representing different oligotrophic characteristics…

We added a section on the anti-estuarine circulation in the last paragraph of the introduction:

The anti-estuarine circulations at the Strait of Gibraltar and the Straits of Sicily, transport low-nutrient surface waters into the basins, and deeper waters out of the basins, resulting in oligotrophic conditions in the western and ultra-oligotrophic conditions in the eastern Mediterranean basin (Krom et al., 2004; Tanhua et al., 2013).

□ Phytoplankton abundance measurements may be underestimated due to wrong preservation of the samples. Freezing the seawater samples in -20 °C rather than in liquid nitrogen and then -80 °C slowly generates ice crystals that may break some of the cells, and thus result in underestimation of the actual counts. Indeed, the pico-phytoplankton cell abundances presented in Figure 5 (and corresponding text) are 1-2 orders of magnitude lower than usually reported in the Mediterranean Sea (~$10^4$-$10^5$ cells/ml).

The cell preservation method using fixation and freezing has been tested in our lab previously. In accordance with previous studies (e.g. Lepesteuer et al., 1993), we used this method and do not expect overproportional cell loss.

It is true that the abundances are quite low at the surface. During the cruise, flow cytometry was also used down to 200 m depth to measure the phytoplankton abundance, showing that phytoplankton numbers were generally higher at 5-100 m than in the SML and ULW which makes it difficult to compare the general surface waters to the SML abundances. Since 5-100 m was beyond the scope of this paper, these numbers were not included in the

manuscript. The numbers retrieved from the SML are in line with previous phytoplankton SML measurements (e.g. Joux et al., 2006).

☐ In section 2.7 you describe how you calculated the 'enrichment factors' between the SML and ULW, however this data is not presented in the manuscript (but only used as a correlation variable). I suggest adding a table with the EF values (and whether the differences were significant or not). It will greatly help the reader to understand the differences between the two water layers.

We have added a table of EFs in the supplementary material (Table S1). The significance of the two different water layers is already described in section 3.2

☐ Figure 5 should be revised. Briefly, there's seems to be problems in the units used for TEP (area and concentration), the panels are not numbered which makes the reading more difficult to follow, the dot's color-code is unclear etc. Please see more details in the table below. Further, I suggest adding to the supporting information a few microscopic images showing example TEP area.

We added numbering to the panels and explained the dot's colour-code. The units used for TEP area and concentration is according to what is commonly used in literature. Information, including pictures on TEP area calculation, can be found in Engel, 2009 (Determination of marine gel particles).

☐ By pre-filtering the seawater onto 100 µm mesh (line 96), you may have removed some fungi and large-size diatoms/dinoflagellates (as indicated in lines 175-177). Please justify this pre-filtration step.

A justification was added in the M&M part:

400 ml of sample was pre-filtered through a mesh with 100 µm pore size in order to avoid zooplankton being captured on the filters and dominating the retrieved 18S sequences and subsequently filtered onto a Durapore membrane (Millipore, 47 mm, 0.2 µm) and immediately stored at -80°C.

And the pre-filtration is also mentioned in the discussion:

Even though diatoms most likely were not dominant in the samples, finding no diatom orders over 1 % in at least one of the samples might also indicate a bias of the primers used or of the pre-filtration removing larger cells and aggregates.

☐ I think the discussion should be elaborated. For example:

❖ You should discuss why you don't see any differences in the eukaryotic diversity between the SML and underlying water in all sites (it's not the organic matter…). Given the lack of (spatial) correlation between phytoplankton and microgels/ TCHO may infer that these organic matter may be refractory in the eastern basin compared to the western basin, or that phytoplankton/bacteria are outcompeted for these 'goods' (perhaps the fungi?).

We have added to the existing discussion:

Bacterial and microalgal numbers determined by flow cytometry decreased significantly from west to east, with bacteria showing the greatest decline. Even though overall bacterial numbers decrease, further molecular analyses would be needed to determine if the bacterial community is changing from west to east and if certain bacterial taxa can benefit from the ultra oligotrophic conditions.

❖ I suggest you discuss how fungi may interact with phytoplankton and bacteria in marine LNLC environments. Do they utilize the same nutrients (thereby competing with the microbes)? Did you find any toxic fungi in the different layers? Can you say anything about the role of fungi in the SML and ULW's food web? Please provide information on fungi biomass/ activity and diversity in other LNLC regions. Do you expect that fungi be more important in oligotrophic vs. meso-/eutrophic marine environments? Does your findings comparable to these other sites?

We have added a description on the ecological role of fungi in the marine environment in general and a more specific description of fungi in the SML.

Fungi can either live a saprotrophic lifestyle, potentially competing for nutrients and substrate with phytoplankton or bacteria and adding to the recycling of organic matter, or they can live a parasitic lifestyle, directly attacking phytoplankton and potentially altering phytoplankton community composition through selective parasitism (Amend et al., 2019; Grossart et al., 2019). By parasitizing phytoplankton, fungi can render inedible phytoplankton accessible for zooplankton (Grossart et al., 2019). Coastal myconeuston in the English Channel was dominated by Ascomycota and Basidiomycota classes (Taylor and Cunliffe, 2014) compared to Ascomycota dominating the surface samples in the current study.

Minor comments/suggestions

- Section 2.1 It is unclear how much water were collected in each station, what were the collection hours (day vs. night. . .), and how much time it took to collect it. Seems that for all analyses the authors needed ~0.5 L from the SML, which is a lot when using the glass plate approach (Harvey, 1966). . . More importantly, if samples were collected in different hours of the day (e.g., S7 vs. S3 based on irradiance presented in Table 1), this might affect the microbial communities in the SML and ULW through daily migration of zooplankton and thus gazing. This issue can affect the abundance/diversity of the eukaryotic microbes in both water layers. Please provide the information and discussion where appropriate. Did you use a mechanic instrument (that can also control the sampling rate; ~17 cm s-1)? If so, an image showing this instrumentation may be a nice addition (especially given that the link you provided of Cunliffe and Wurl, 2014 does not work. . .).
  We added information on sampling time, duration and volume in the M&M and table 1:

  A total of app. 1.5 L of SML sample was collected in the course of 1 h. Sampling times are listed in table 1.

  As mentioned in the results section 3.2, influence of irradiation was tested, but no significant correlations were found, making day/night sampling effects unlikely.

  As described in the M&M, the sampling was carried out manually with a standard glass plate, counting the timing of the dips resulting in an average sampling rate of ~17 cm s-1.

  We thank the reviewer for his remark and have updated the link to Cunliffe and Wurl, 2014.

- Line 57 Pourquoi pas? or Pourquoi pas
  The official name of the research vessel is Pourquoi pas?

- Line 68 ". . .The abundance and area of TEP and was measured microscopically. . ." (removed "and")
  We removed the 'and'

- Line 77 (ditto line 83) Freezing seawater samples in -20 ∘C without pre-freezing it in liquid nitrogen may result in cell lose. The slow freezing at -20 ∘C creates ice crystals which results in cell breakdown, thus leading to underestimation of pico-phytoplankton/bacterial abundances. Indeed – your pico-phytoplankton cell abundance (e.g., Figure 5) is low by 1-2 orders of magnitude relative to previous studies from the Mediterranean surface water. Do the numbers presented in Figure 5 only show the eukaryotic algae (i.e., without the cyanobacteria)?
  The cell preservation method using fixation and freezing has been tested in our lab previously. In accordance with previous studies (e.g. Lepesteuer et al., 1993), we used this method and do not expect overproportional cell loss.

  It is true that the abundances are quite low at the surface. During the cruise, flow cytometry was also used down to 200 m depth to measure the phytoplankton abundance, showing that phytoplankton numbers were generally higher at 5-100 m than in the SML and ULW which makes it difficult to compare the general surface waters to the SML abundances. Since 5-100 m was beyond the scope of

this paper, these numbers were not included in the manuscript. The numbers retrieved from the SML are in line with previous phytoplankton SML measurements (e.g. Joux et al., 2006).

The numbers represented in figure 5 show all algae, including cyanobacteria.

- Abstract, Line 82 and throughout Your flow-cytometry analyses enabled you to enumerate pico/nano-phytoplankton and not the total phytoplankton fraction which is also comprised of larger algal communities (large diatoms, dinoflagellates etc.). What is the cell-size range of the flow-cytometer you used? Usually, to get total "phytoplankton" you should have measured chlorophyll.a and /or run complimentary microscope analyses. This is especially important given that the SML is rich in large-size phytoplankton (Hardyet al., 1988). I suggest changing the term "phytoplankton" to "picophytoplankton" throughout. BTW – Did you see any differences (SML vs. underlying water and between basins) in the pico-phytoplankton communities (e.g., Prochlorococcus:Synechococcus ratio, prokaryotes:eukaryotes ratio)?

Prochloroccocus were not detected in the flow cytometer. According to the suggestion, we changed the phytoplankton detected in the flow cytometer to picophytoplankton as the size of cells detected in the instrument is app. 0.2 – 20 µm.

- Paragraph in lines 42-49 I suggest to remove from the introduction (and maybe move it to the discussion?). I'm aware you tried to describe what is TEP, but it has little connection the way it's written with the SML's background. Maybe adding a sentence saying that TEP prevalent the SML. . . etc.

We added an explanation on why TEP are important in the SML in the introduction:

TEP contain mainly polysaccharides (Mopper et al., 1995; Passow, 2002), occur ubiquitously in the ocean (Alldredge et al., 1993; Passow, 2002; Engel et al 2020), and are an important structural component of the SML (Wurl and Holmes, 2008).

- Line 50 "looked at the spatial distribution. . ." (add "the") Figure 5 âA̧ c Please number the different ́ panels (A, B. . .) and revise the legends accordingly ("Abundance of bacteria (A), pico-phytoplankton (B), TEP area (C). . ."). âA̧ c What's the difference between the ́ gray and black dots (different cruises?)? âA̧ c TEP area – I don't understand the ́ units. What is mm L-1 (ditto in the text)? I suggest adding a figure in the SI (or atl least in your reply) explaining this. âA̧ c TEP concentration – I don't understand the ́ units. what is TEP per L-1 (ditto in the text)? Do you mean µg GX L-1 (if so, there's something off with the numbers).

We thank the reviewer for the comment and have added panels in figure 5. TEP area in mm2 L-1 and TEP L-1 are commonly used in the literature to describe TEP which was measured microscopically, whereas µg GX L-1 is used for TEP measured colorimetrically (see also Engel et al., 2020).

Please see below an example of TEP area measurement from Engel, 2004 a – All TEP are marked using a manual threshold in ImageJ, then the program automatically calculates the area that is taken up by TEP.

[Figure]

TEP L-1 is describing, how many particles are found per litre of seawater (using the above pictures, instead of taking the area into account, only counting how many particles were found per field of view).

- Line 96 and lines 175-177 Please justify why you used pre-filtration for the DNA extractions. By doing so, you may have taken out fungi's mycelium as well as large-size diatoms/dinoflagellates (that are often found in the western basin water as indicated in lines 175-177). You may have also taken out TEP with its rich microbiome (algae, bacteria and fungi).

A justification was added in the M&M part:

400 ml of sample was pre-filtered through a mesh with 100 µm pore size in order to avoid zooplankton being captured on the filters and dominating the retrieved 18S sequences and subsequently filtered onto a Durapore membrane (Millipore, 47 mm, 0.2 µm) and immediately stored at -80°C.

And the pre-filtration is also mentioned in the discussion:

Even though diatoms most likely were not dominant in the samples, finding no diatom orders over 1 % in at least one of the samples might also indicate a bias of the primers used or of the pre-filtration removing larger cells and aggregates.

Regarding the removal of TEP: While some very large TEP might have been removed, the vast majority of TEP that were found on the TEP filters (which were not pre-filtered) were less than 100 um in diameter.

- Section 4.1 You should discuss why you didn't see any differences between the SML and underlying water in all sites, while chemically-wise (total carbohydrates and TEP) you found significant differences. Currently, the discussion in this section is a bit weak.
  We expanded the discussion:

However, looking at the current study, no inhibition by UV radiation can be inferred from the data because phytoplankton were enriched despite high radiation values (e.g. stations S4 and 7) (Table 1). At the same time, TEP were significantly enriched during the sampling campaign while the phytoplankton community did not show significant differences. Previous studies suggested that TEP can protect phytoplankton and bacteria from UV radiation (Elasri and Miller, 1999; Ortega-Retuerta et al., 2009). Further studies would be needed to determine whether TEP production was higher in the SML due to phytoneuston UV protection or whether TEP formation rates were higher in the SML due to wind and wave shear at the surface (Carlson, 1993; Cunliffe et al., 2013).

- Lines 195-197 There's also a possibility it's a contamination. . . Did you run blank filters? You can also look at the air-mass backward trajectories (https://ready.arl.noaa.gov/HYSPLIT_traj.php) and see where aerosols came from the day before you sampled there, namely if you received any terrestrial origin particles.

  Blank filters were not run daily, however, contamination of land plants seems unlikely since there were no plants in the lab or near the CTD sampling station onboard the RV.

  We thank the reviewer for suggesting the website. Please find below the output which suggests that the aerosols came from a terrestrial input. We have also included the below graph in the supplementary information and added an additional short explanation in the discussion:

  This is also corroborated by air mass trajectory backtracking using the HYSPLIT model (Fis. S3) which shows that aerosols likely were of terrestrial origin.

[Figure]

- Line 217 "very oligotrophic" (instead of "veryoligotrophic"). BTW- I suggest saying 'ultra-oligotrophic'.
  We changed the wording to ultra oligotrophic

- Lines 218-219 Please cite a reference to back up this statement.
  We apologise for the confusing phrasing, we were talking about our study and changed the sentence to:

  This is in good accordance with the present study in the ultra oligotrophic eastern Mediterranean Sea (Durrieu de Madron et al., 2011; Fogg, 1995; Wikner and Hagstrom, 1988) where low phytoplankton abundances, but high TEP enrichments of 1.1-17.3 were found in the present study.

**Reply reviewer #2**

Review of "Eukaryotic community composition in the sea surface microlayer across an east-west transect in the Mediterranean Sea" This is an interesting work reporting original data on the abundance of eukaryotes and TEPs in a East-West transect in the Mediterranean. The data set seems of good quality, even though the depth of the discussion could be improved in some sections. I recommend the following modifications:

We thank the reviewer for the helpful comments. Please find below our reply to the specific comments.

-Line28. There are some reports on bacteria in the SML, also some recent work using 16S. I suggest this could be cited here.

Thank you for the suggestion, we added this in the introduction:

Despite the fact that microbes in the SML can directly and indirectly influence air-sea gas exchange, few studies have looked at the microbial community composition in the SML, mainly focussing on bacteria (Agogué et al., 2005; Joux et al., 2006; Obernosterer et al., 2008) and less on microbial eukaryotes (Taylor and Cunliffe, 2014).

- Lines 43-49. This paragraph is un-related with the rest of the introduction and the title of the manuscript. It could be removed from the introduction.

We have removed the paragraph.

- Line 53. It is not clear to me to which degree the organic matter in the SML is related to atmospheric inputs, it may be driven by partitioning of surfactant-like chemicals from underlying waters or exudates from microbes. In addition, later in the text it is said that the atmospheric influence is covered in another manuscript in preparation.

We have removed the reference to atmospheric inputs from the sentence:

The present study focuses on the organic matter (OM) and microbial eukaryotes distribution, focusing on the myconeuston community composition in the SML of the Mediterranean Sea using samples collected during the PEACETIME cruise in May and June 2017.

- Lines 63. I find it difficult to believe that such a control of this rate can be achieve, especially on a moving zodiac.

We have removed the specific speed:

The glass plate was immersed and withdrawn slowly and perpendicular to the sea surface.

- Line 65. How was it collected?

We expanded the explanation on the ULW collection:

The ULW samples were collected concurrently with two acid-cleaned and MilliQ rinsed glass bottles by immersing the closed bottles and opening them at app. 20 cm.

- Volume of SML and ULW collected?

We expanded the description of sample collection:

A total of app. 1.5 L of SML sample was collected in the course of 1 h. Sampling times are listed in table 1.

- Line 131. Is this the result of a spearman correlation? R=0.1 means R2=0.01. Even though p<0.05, I don't think this shows a high similarity. I think it needs an explanation of which data was used for this similarity measure.

We used ANOSIM to compare the eukaryotic communities in the SML and the ULW on ranked data based on Bray-Curtis dissimilarity. A R value close to 1 suggests dissimilarity between groups, whereas an R value close to 0 suggests no dissimilarity between groups, in our case no dissimilarity between the SML and the ULW.

- ASV is never defined in the manuscript, I guess it is "amplicon sequence variant", but a definition, and probably an explanation, is needed in methods or the first time it appears.

Yes, we meant amplicon sequence variant and have added the explanation to the first time ASV is appearing in the text.

- Were differences between SML and ULW tested with a paired test?

The differences between SML and ULW for flow cytometry counts and organic matter concentrations were determined with a t-test on the enrichment factors (EFs) over the different stations, thus there was no need for a paired test, since through the calculation of the EF the only respective SML and ULW samples were compared.

- Generally, the EF could be correlated with environmental variables such as wind speed.

That is a good point and we have correlated environmental variables with the EF, as described below for TEP: While irradiation, water temperature and salinity did not correlate with TEP abundance or area, wind speed did have a significant negative correlation with TEP abundance in the SML ($R^2$ = -0.73) and TEP area in the SML ($R^2$ = -0.75) and the enrichment factor of TEP area ($R^2$ = -0.63).

- Was DOC measured? This could influence the east-west differences, as well their composition.

While DOC could give interesting additional information, it was not measured in the SML throughout the cruise.

- Line 153. Rewrite...

We have rephrased the sentence to make it clearer:

Picophytoneuston abundance was on average 3.3 x $10^3$ ± 1.9 x $10^3$ cells ml$^{-1}$ in the SML and picophytoplankton abundance in the ULW was on average 2.3 x $10^3$ ± 1.7 x $10^3$ cells ml$^{-1}$ (range of 1.4 x $10^3$ – 8.5 x $10^3$ cells ml$^{-1}$ in the SML, 9.5 x $10^2$ – 7.1 x $10^3$ cells ml$^{-1}$ in the ULW).

- Line 176. Couldn't diatoms and other phytoplankton groups be very affected by UV radiation?, ok, this is commented later, but then the enrichment of phytoplankton can be derived by physical processes (buoyancy and fractionation at surface due to surface tension related issues), then this would be independent of radiation.

We included buoyancy of cells as a potential community shaping factor in the discussion:

In addition to buoyancy of cells, radiation, especially in the SML, where often high levels of UV-radiation occur, could potentially cause damage by photoinhibition.

- Note that the SML is generally enriched in hydrophobic and surfactant-like chemicals as many anthropogenic compounds. This has been described for PAHs, alkanes, PCBs, Perfluoralkyl substance, etc in the Mediterranean and elsewhere. This could also have an influence on the east-west differences, as concentrations of POPs in biota are higher in oligotrophic regions due to a complex interplay of factors (Berrojalbiz et al. 2011, Morales et al. 2015, González-Gaya et al. 2019). For example, for bacteria, it has been shown that the SML is especially enriched in those taxa having the potential to degrade pollutants (Martinez-Varela et al. 2020). Even though, organic pollutants were probably not measured in this cruise, it could be another factor to take into account or comment shortly. Fungi are known as being very efficient degrading persistent pollutants.

We thank the reviewer for this comment and have expanded the discussion to include the possible influence of chemicals enriched in the SML:

In addition, previous studies have shown that the Eastern Mediterranean Sea shows higher concentrations of organic pollutants (Berrojalbiz et al., 2011a, 2011b) and a Cladosporium strain has been observed to degrade polycyclic aromatic hydrocarbons (Birolli et al., 2018), highlighting another potential substrate for the fungi detected in the Ionian Sea.

- Line 231. Here and in other parts of the manuscripts. Bacteria was determined by bacterial counts (abundances), but this is a very limited information for this types of statements, as even bacterial abundance decrease, the abundance of some key taxa may increase.

We have added a sentence in the discussion on this limitation:

Even though overall bacterial numbers decrease, further molecular analyses would be needed to determine if the bacterial community is changing from west to east and if certain bacterial taxa can benefit from the ultra oligotrophic conditions.

**Reply reviewer #3**

General comments: The MS by Zäncker and colleagues reports data on autotrophic and heterotrophic microbial cell abundances, TEP, carbohydrates, and 18S sequences from water collected in the sea surface microlayer (SML) and in underlying waters (ULW) in different basins in the Mediterranean Sea. The work presents a partial description of the biological and chemical characterization of the SML and ULW.

We thank reviewer #3 for reviewing the manuscript and the comments. Please find below a detailed answer to the raised questions and issues:

However, the way the data are presented and in particular discussed leaves the question on the authors' specific aim(s) open. The question that arises is 'What is the link between TEP, carbohydrates and microbial communities?' The authors provide no rationale for combining these specific results in one MS.

Carbohydrates are precursors for TEP (as stated in the introduction, line 42: Phytoplankton and phytoneuston can release precursors such as carbohydrates which can aggregate and form gelatinous particles such as transparent exopolymer particles (TEP). (Chin et al., 1998; Engel et al., 2004; Verdugo et al., 2004). TEP contain mainly polysaccharides (Mopper et al., 1995; Passow, 2002), occur ubiquitously in the ocean (Alldredge et al., 1993; Passow, 2002), and are an important structural component of the SML (Wurl and Holmes, 2008).), thus looking at TEP and carbohydrates combined provides a more complete picture than simply TEP alone. Since TEP are valuable for microbes as attachment site and food source and are structurally crucial for the SML, which is the target region of the present study, the authors concluded that all three components (carbohydrates, TEP, eukaryotes) are important components of the study and should thus be included. We have added this explanation also in the introduction:

The present study focuses on TEP as important structural components of the SML and their precursors, carbohydrates, as well as microbial eukaryotes distribution, focusing on the myconeuston community composition in the SML of the Mediterranean Sea using samples collected during the PEACETIME cruise in May and June 2017.

Further, the results are compared between SML and UW as well as among basins, which adds another level of complexity.

The authors acknowledge the added level of complexity, but given the major differences in not only eukaryotic community composition, but also trophic status and exchange with Atlantic waters in the different basins, when treating all basins together a lot of variability in the data would be lost.

The discussion of the data (in particular in the context of atmospheric deposition) is difficult to follow.

We have addressed more specific points on the atmospheric deposition below. In addition, we have used the NOAA HYSPLIT model to show the backwards trajectory of air masses 2 days prior to sampling in the Ionian Sea. The model results show that the air masses very likely originated above land, further corroborating the idea that the fungi found in the Ionian Sea, while thriving in this area of the Mediterranean Sea, have been introduced from terrestrial sources.

[Figure]

NOAA HYSPLIT MODEL
Backward trajectories ending at 0900 UTC 23 May 17
GDAS Meteorological Data

Job ID: 1585          Job Start: Mon Sep 14 09:33:10 UTC 2020
Source 1 lat.: 37.983000 lon.: 17.625000 heights: 500, 1000 m AGL

Trajectory Direction: Backward     Duration: 48 hrs
Vertical Motion Calculation Method:     Model Vertical Velocity
Meteorology: 0000Z 22 May 2017 - GDAS1

This MS is a contribution to the Special Issue of the PEACETIME project. The characteristics of the SML will certainly provide important insights to the overall project. I consider, however, that the MS cannot be accepted in its present form, but needs major revisions. I advice the authors to re-consider their main objective(s) and to present only the appropriate data. My further suggestion is to re-construct the discussion in a way that it focuses on the data presented in this MS. An original finding of the study is the high relative abundance of fungi sequences in the Thyrennian Sea, both in the SML and ULW. One possibility would be to focus the MS on eukaryotic diversity and fungi in particular.

As stated above, the authors have carefully considered which data to include in the manuscript and feel that including carbohydrates, TEP and microbes gives the best possible overview of SML dynamics.

We considered the suggestion of the reviewer to restructure the discussion and agree that the high relative abundance of fungi sequences in the Ionian Sea is an original and interesting finding. However, the discussion is already focussed on eukaryotic diversity (section 4.1 Eukaryotic diversity in the surface of the Mediterranean Sea, lines 167-187) and fungi in particular (section 4.2 Fungi in

the Ionian Sea, lines 189 – 231), and thus the authors feel like a restructuring of the discussion is not meaningful at this point.

Specific comments:

Abstract: Line 10-11: One understands that this main objective of the work, but it is not focus of the following sections.

Changed it to:

However, little is known about the distribution of microbial eukaryotes in the SML.

Methods: Line 89: If I understand correctly, the 20 mL samples collected onboard and frozen (-20∘) were not ultra filtered. The ultrafiltration step was done in the lab. I suggest to clarify this.

We clarified this:

In the home lab, high performance anion exchange chromatography with pulsed amperometric detection (HPAEC-PAD) was applied on a Dionex ICS 3000 ion chromatography system (Engel and Händel 2011) for TCHO analysis.

Results: Line 150: Please refer to Fig. 5 in the text

We added a reference to figure 5.

Line 153-155 and legend of Fig. 5: Flow cytometry was used in the present study to determine phytoplankton abundance in the SML and in ULW. I suggest the authors clarify here the size fraction of the organisms that can be determined by flow cytometry (i.e. generally up to 20 µm). Any larger phytoplankton are not included in their counts.

We named the phytoplankton measured in the flow cytometer picophytoplankton and included the size range (0.2 – 20 µm) in the results section.

Line160: should be Fig. 5

Thanks for pointing this out, we have changed it accordingly.

Discussion: Line 173: Sequencing data provide information on the relative abundance of a given taxonomic unit, but no absolute values. I suggest re-writing the sentence accordingly.

We changed 'concentrations' to 'relative abundances'.

Line 191: In the previous paragraph the authors discuss the potential biases of sequencing data due to differences in gene copy numbers, and I totally agree. How would this impact their observations on fungi sequences? I suggest the authors include a short description of what is known on fungi copy numbers and whether this could have led to a potential overestimation in their data set.

We added a sentence on the impact on fungal sequences:

While fungi, like dinoflagellates and other eukaryotic groups, can have varying amounts of 18S rDNA gene copy numbers, the patchy distribution of fungi found in this study makes a consistent bias unlikely.

Line 194 and elsewhere: Please apply the term 'relative abundance' instead of 'abundance' or 'amount' (line 203)

We have made changes throughout the manuscript accordingly.

Line 198: What is the rationale for the conclusion that dust and rain lead to an increase in TEP and in unidentified dinoflagellates in the SML? Even if another MS on this issue is in preparation, a more information is required here for an appropriate discussion.

We have changed the wording of the sentence to 'coincided' instead of 'resulted' to not make claims that we don't discuss in the MS.

Station FAST_2 in the western basin was highly influenced by dust input in the area (Guieu et al., 2020; Tovar-Sánchez et al., 2020). This coincided not only with a high increase in TEP abundance in the SML, but also with a distinct increase in the relative abundance of unidentified dinoflagellates in the SML (Fig. 3).

What is meant by 'previous to the research campaign'? A few days or weeks? Where is this shown in Fig. 3?

Station FAST_2 is represented by the second bar in figure 3. We have changed the sentence to be more specific about the timing:

either deposited by dust or by rain days before this research campaign in the Ionian Sea or in other areas closeby.

Line 201: As mentioned above, this discussion does not refer to any data presented in this MS, and thus confusing.

As stated in the text, the community data is indeed presented in figure 4 and helps to rule out dust input as the main influencing factor of eukaryotic community composition. Thus, the authors feel like this section should be kept in the manuscript.

Line 202-203: It seems the authors contradict their statements above. Please clarify.

We have changed the sentence to make our point better understandable:

In addition, the highest relative abundance of fungi was found in the ULW and not the SML, making a simple atmospheric influence without any subsequent thriving of certain fungal taxa unlikely.

**Reply editor**

Comments to the Author:

Thank you for your answers to the 3 reviewers that mostly covers the issues raised. It is also nice that additional information are provided as SI (air masses backwards trajectories). However I have few additional remarks that I'd like you take into account before the paper can be accepted for publication:

We thank the editor for her useful comments. Please see below our specific replies to the issues raised.

- I found it unfortunate that only few papers resulting from the PEACETIME cruise are cited. I would recommend at least to add Sellegri et al. (accepted Scientific reports) that you co-authored that nicely emphasize on the importance of SML in air-sea exchange processes, this is directly link to your work during PEACETIME.

We've added citations to Sellegri et al., Freney et al., Fu et al., Desboeufs et al., Trueblood et al. in the manuscript.

- L56, please add following references : MERMEX Group, 2011 and Pujo-Pay et al., 2011 (PUJO-PAY, Mireille, CONAN, Pascal, ORIOL, L., et al. Integrated survey of elemental

stoichiometry (C, N, P) from the western to eastern Mediterranean Sea. Biogeosciences, 2011, vol. 8, no 4, p. 883-899.)

We have added the citations.

(note that De Madron et al. should be MERMEX Group et al.: in the text and references)

We changed the citation and reference accordingly.

- the answer to several remarks is relevant but is not always reflected in the text: I think it is important that your answers (and the new references discussed/proposed to support your remarks) are properly included in your manuscript.

In particular, rem 3 rev#1 (preservation methodology that could impact the abundance (that is lower than usually reported in the Med Sea)

We have added more explanations:

Line 95: This method is in accordance with previous method development (Lepesteur et al., 1993).

Line 241-244: While overall microalgal abundances determined by flow cytometry were rather low in the SML and ULW, they were comparable to other studies looking at the phytoplankton abundance in the SML of the Mediterranean Sea (Joux et al., 2006). The microalgal numbers from 5-200 m (data not shown) were higher than at the air-sea interface.

also from rev#2 : there are some DOC data although not measured in all SML samples and it would be interesting to check the relevance of this parameter and indicate weither or not how the differences impact the east-west pattern ?

We added a short discussion of DOC and POC (as reported by Freney et al. and Trueblood et al.) in the discussion (Lines 306-308):

At the same time, TCHO and TEP were still abundant in the Ionian Sea, as well as DOC in the SML and DOC and POC in the ULW which did not show changes between the Ionian Sea and the other basins (Freney et al., 2020; Trueblood et al., 2020).

For clarification please find below the figure this sentence is referring to:

[Figure]

**Figure S2. Various biogeochemical measurements in the SML (left) and SSW (right).**

also from rev#2, about the bacterial abundance decrease, it would be nice to refer to Tovar-Sanchez et al., 2020.

We added the citation.

- L218, As no dust deposition was evidenced at ION from atmospheric studies (Desboeufs et al. in prep and Fu et al., in prep) and also because dust is often a wet deposition, I suggest to modify the sentence : "...either deposited by dust or by rain days before this research campaign in the Ionian Sea or in other areas closeby" by "... linked to wet or dry deposition that occurred before and/or during our sampling period at ION".

We changed the sentence accordingly.

- L220, please add reference to Fu et al. paper in prep : after the sentence : « This is also corroborated by air mass trajectory backtracking using the HYSPLIT model (Fig. S3) which showed that aerosols likely were of terrestrial origin » (please replace « terrestrial » by continental) and add : « this was confirmed by atmospheric measurements indicating that chemical composition of dry and wet depositions were influenced by Eastern Europe air masses (Desboeufs et al., this issue, in prep.). [Desboeufs, K., Doussin, J.-F., Giorio, C., Triquet, S., Fu, Y., Dulac, F., Garcia-Nieto, D., Chazette, P., Féron, A., Formenti, P., Gaimoz, C., Maisonneuve, F., Riffault, V., Saiz-Lopez, A., Siour, G., Zapf, P., and Guieu, C.: ProcEss

studies at the Air-sEa Interface after dust deposition in the MEditerranean sea (PEAcEtIME) cruise: Atmospheric overview, Biogeosciences this special issue, in preparation.]

We changed the sentence accordingly and added the citations.

- - abstract: as this is important regarding the objectives of the PEACETIME project, I would add L 14 "At the stations located in the Ionian Sea, fungi – ***likely of continental origin via atmospheric deposition*** -were found in high relative abundances etc."

We added the insert accordingly.

I suggest that you reconsider your response for the following remark as they do not fully answer the question raised :

- rev#1 : the second suggestion : appart from the litterature review that you added, the reviewer was wondering how your data in oligotrophic system compare with previous work, this should be done in the text.

☐☐I suggest you discuss how fungi may interact with phytoplankton and bacteria in marine LNLC environments. Do they utilize the same nutrients (thereby competing with the microbes)? Did you find any toxic fungi in the different layers? Can you say anything about the role of fungi in the SML and ULW's food web? Please provide information on fungi biomass/ activity and diversity in other LNLC regions. Do you expect that fungi be more important in oligotrophic vs. meso-/eutrophic marine environments? Does your findings comparable to these other sites?

We have added a description on the ecological role of fungi in the marine environment in general and a more specific description of fungi in the SML and on fungi in other LNLC regions (Lines 215-219):.

Marine fungi can live a saprotrophic lifestyle, degrading and recycling high molecular weight organic matter (Chrismas and Cunliffe, 2020; Cunliffe et al., 2017) and potentially competing with functionally similar bacteria. Some marine fungi are also phytoplankton parasites, potentially altering phytoplankton community composition through selective parasitism (Amend et al., 2019; Grossart et al., 2019).. At present, we have a very limited understanding of diversity and functional role of fungi in the SML (myconeuston). One previous study of the coastal myconeuston in the Western English Channel off Plymouth (UK) showed that the SML was dominated by both Ascomycota and Basidiomycota (Taylor and Cunliffe, 2014), compared to Ascomycota dominating in this study.

So far, not many studies have looked at fungi in LNLC regions. A global comparison of fungal distribution (Hassett et al., 2020) has found that fungal diversity determined by amplicon sequencing varies between different oceanic regions with *Exophiala*, belonging to Ascomycota, dominating the Ligurian Sea samples and an unclassified Ascomycota being the most abundant taxon, similar to our study being dominated by Ascomycota.

**List of all relevant changes**

**Lines 45-47:** TEP contain mainly polysaccharides (Mopper et al., 1995; Passow, 2002),  occur ubiquitously in the ocean (Alldredge et al., 1993; Passow, 2002), and are an important structural component of the SML (Wurl and Holmes, 2008).

**Lines 54-60:** (Durrieu de Madron et al., 2011). The anti-estuarine circulations at the Strait of Gibraltar and the Straits of Sicily, transport low-nutrient surface waters into the basins, and deeper waters out of the basins, resulting in oligotrophic conditions in the western and ultra-oligotrophic conditions in the eastern Mediterranean basin (Krom et al., 2004; Mermex Group et al., 2011; Pujo-Pay et al., 2011; Tanhua et al., 2013). The present study focuses on TEP as important structural components of the SML and their precursors, carbohydrates, as well as  microbial eukaryotes distribution , focusing on the myconeuston community composition in the SML of the Mediterranean Sea using samples collected during the PEACETIME cruise in May and June 2017.

**Lines 68-74:** The glass plate was immersed and withdrawn perpendicular to the sea surface ~~at a controlled rate of ~17 cm s⁻¹~~. With a Teflon wiper, SML samples were collected in acid cleaned and rinsed bottles (Cunliffe and Wurl, 2014). A total of app. 1.5 L of SML sample was collected in the course of 1 h. Sampling times are listed in table 1. All sampling equipment was acid-cleaned (10 % HCl), rinsed with Milli-Q and copiously rinsed with seawater from the respective depth once the sampling site was reached. The ULW samples were collected concurrently with two acid-cleaned and MilliQ rinsed glass bottles by immersing the closed bottles and opening them at app. 20 cm.

**Line 95:** This method is in accordance with previous method development (Lepesteur et al., 1993).

**Lines 104-106:** 400 ml of sample was pre-filtered through a mesh with 100 µm pore size in order to avoid zooplankton being captured on the filters and dominating the retrieved 18S sequences and subsequently filtered onto a Durapore membrane (Millipore, 47 mm, 0.2 µm) and immediately stored at -80°C.

**Lines 163-165:** Picophytoneuston (0.2 – 20 µm size range) abundance was on average 3.3 x $10^3$ ± 1.9 x $10^3$ cells ml⁻¹ in the SML and picophytoplankton abundance in the ULW was on average 2.3 x $10^3$ ± 1.7 x $10^3$ cells ml⁻¹ (range of 1.4 x $10^3$ – 8.5 x $10^3$ cells ml⁻¹ in the SML, 9.5 x $10^2$ – 7.1 x $10^3$ cells ml⁻¹ in the ULW).

**Lines 184-185:** Looking at the phytoplankton community (Fig. 3), it becomes apparent that no diatoms were present at high relative abundances.

**Lines 187-188:** Even though diatoms most likely were not dominant in the samples, finding no diatom orders over 1 % in at least one of the samples might also indicate a bias of the primers used or of the pre-filtration removing larger cells and aggregates.

Lines 198-202: At the same time, TEP were significantly enriched during the sampling campaign while the phytoplankton community did not show significant differences. Previous studies suggested that TEP can protect phytoplankton and bacteria from UV radiation (Elasri and Miller, 1999; Ortega-Retuerta et al., 2009). Further studies would be needed to determine whether TEP production was higher in the SML due to phytoneuston UV protection or whether TEP formation rates were higher in the SML due to wind and wave shear at the surface (Carlson, 1993; Cunliffe et al., 2013).

Lines 208-220: While fungi, like dinoflagellates and other eukaryotic groups, can have varying amounts of 18S rDNA gene copy numbers, the patchy distribution of fungi found in this study makes a consistent bias unlikely. Marine fungi can live a saprotrophic lifestyle, degrading and recycling high molecular weight organic matter (Chrismas and Cunliffe, 2020; Cunliffe et al., 2017) and potentially competing with functionally similar bacteria. Some marine fungi are also phytoplankton parasites, potentially altering phytoplankton community composition through selective parasitism (Amend et al., 2019; Grossart et al., 2019).. At present, we have a very limited understanding of diversity and functional role of fungi in the SML (myconeuston). One previous study of the coastal myconeuston in the Western English Channel off Plymouth (UK) showed that the SML was dominated by both Ascomycota and Basidiomycota (Taylor and Cunliffe, 2014), compared to Ascomycota dominating in this study.
So far, not many studies have looked at fungi in LNLC regions. A global comparison of fungal distribution (Hassett et al., 2020) has found that fungal diversity determined by amplicon sequencing varies between different oceanic regions with *Exophiala*, belonging to Ascomycota, dominating the Ligurian Sea samples and an unclassified Ascomycota being the most abundant taxon, similar to our study being dominated by Ascomycota.

Lines 221-228: Not only fungal relative abundances increased in the Ionian Sea, but also ASVs identified as Solanales (Nicotiana) had quite high relative abundances in the easternmost stations. Since Solanales are land plants, presence of their DNA could suggest a possible strong terrestrial influence on the Ionian Sea, linked to wet or dry deposition that occurred before and/or during our sampling period at ION. either deposited by dust or by rain previous to this research campaign in the Ionian Sea or in other areas closeby. This is also corroborated by air mass trajectory backtracking using the HYSPLIT model (Fisg. S3) which showed that aerosols likely were of terrestrialcontinental origin (Fu et al., in prep), also confirmed by atmospheric measurements indicating that chemical composition of dry and wet depositions were influenced by Eastern European air masses (Desboeufs et al., in prep.n.d.).

Lines 245-253: Bacterial and microalgal numbers determined by flow cytometry decreased significantly from west to east, with bacteria showing the greatest decline (Tovar-Sánchez et al., 2020). While overall microalgal abundances determined by flow cytometry were rather low in the SML and ULW, they were comparable to other studies looking at the phytoplankton abundance in the SML of the Mediterranean Sea (Joux et al., 2006). The microalgal numbers from 5-200 m (data not shown) were higher than at the air-sea interface. Even though overall bacterial numbers decrease, further molecular analyses would be needed to determine if the bacterial community is changing from west to east and if certain bacterial taxa can benefit from the ultra oligotrophic conditions. At the same time, TCHO and TEP were still abundant in the Ionian Sea, as well as DOC in the SML

and DOC and POC in the ULW which did not show changes between the Ionian Sea and the other basins (Freney et al., 2020; Trueblood et al., 2020).

[revised manuscript text omitted]